# Dynamic influence maximization

**Binghui Peng**
Columbia University
bp2601@columbia.edu

## Abstract

We initiate a systematic study on *dynamic influence maximization* (DIM). In the DIM problem, one maintains a seed set $S$ of at most $k$ nodes in a dynamically involving social network, with the goal of maximizing the expected influence spread while minimizing the amortized updating cost. We consider two evolution models. In the *incremental model*, the social network gets enlarged over time and one only introduces new users and establishes new social links, we design an algorithm that achieves $(1-1/e-\epsilon)$-approximation to the optimal solution and has $k \cdot \text{poly}(\log n, \epsilon^{-1})$ amortized running time, which matches the state-of-art offline algorithm with only poly-logarithmic overhead. In the fully dynamic model, users join in and leave, influence propagation gets strengthened or weakened in real time, we prove that under the Strong Exponential Time Hypothesis (SETH), no algorithm can achieve $2^{-(\log n)^{1-o(1)}}$-approximation unless the amortized running time is $n^{1-o(1)}$. On the technical side, we exploit novel adaptive sampling approaches that reduce DIM to the dynamic MAX-k coverage problem, and design an efficient $(1-1/e-\epsilon)$-approximation algorithm for it. Our lower bound leverages the recent developed distributed PCP framework.

## 1 Introduction

Influence maximization (IM) is the algorithmic task of given a social network and a stochastic diffusion model, finding a seed set of at most $k$ nodes with the largest expected influence spread over the network [35]. Influence maximization and its variants have been extensively studied in the literature over past decades [35, 13, 29, 45, 49, 38, 4, 22, 8], and it has applications in viral market, rumor control, advertising, etc.

Social influence can be highly dynamic and the propagation tendencies between users can alter dramatically over time. For example, in a Twitter network, new users join in and existing users drop out in real time, pop stars arise instantly for breaking news and trending topics; in a DBLP network, scientific co-authorship is built up and expands gradually over time. The classic IM algorithms make crucial assumptions on a stationary social network and they fail to capture the elastic nature of social networks. As a consequence, their seeding set could become outdated rapidly in a constantly involving social network. To mitigate the issue, one designs a dynamic influence maximization algorithm, which maintains a feasible seed set with high influence impact over time, and at the same time, saturates low average computation cost per update of the social network.

In this paper, we initiate a systematic study on the dynamic influence maximization (DIM) problem . Two types of evolution models are of interests to us. In an *incremental* model[1], the social networks keep growing: new users join in and new social relationship is built up or strengthened over time. The motivating example is the DBLP network, where co-authorship gets expanded over time. Another justification is the preferential attachment involving model of social network [12, 41]. In a *fully*

---

[1]These terms are standard notions in the dynamic algorithm literature, e.g., see [30, 1]

*dynamic* model, the social network involves over time, users can join in and leave out, social links emerge and disappear, influence impacts get strengthened or weakened. The motivating examples are social media networks like Twitter or Facebook, and the advertising market.

There are some pioneer works on the topic of dynamic influence maximization and various *heuristic* algorithms have been proposed in the literature [52, 44, 24]. These algorithms have different kinds of approximation guarantee on the seed set, but the crucial drawback is that the amortized running time per update is no better than $\Omega(n)$, here $n$ is the total number of nodes in a social network. This is extremely unsatisfactory from a theoretical view, as it indicates these algorithm do no better than re-run the state-of-art IM algorithm upon every update. Hence, the central question over the field is that, *can we achieve the optimal approximation guarantee with no significant overhead on amortized running time?*

**Our contribution**   We address the above questions and provide clear resolutions. In the incremental model, we prove it is possible to maintain a seed set with $(1 - 1/e - \epsilon)$-approximation ratio in amortized running time of $k \cdot \text{poly}(\log n, \epsilon^{-1})$, which matches the state-of-art offline algorithm up to poly-logarithmic factors (Theorem 3.1). While in the fully dynamic algorithm, assuming the Strong Exponential Time Hypothesis (SETH), we prove no algorithm can achieve $2^{-(\log n)^{1-o(1)}}$ approximation unless the amortized running time is $n^{1-o(1)}$ (Theorem 4.3 and Theorem 4.4). This computational barrier draws a sharp separation between these two dynamic models and delivers the following surprising message: There is no hope to achieve any meaningful approximation guarantee for DIM, even one only aims for a slightly reduced running time than the naive approach of re-instantiating an IM algorithm upon every update.

On the technical side, our DIM algorithm in the incremental model uses novel adaptive sampling strategies and reduces the DIM problem to the dynamic MAX-k coverage problem. We provide an efficient dynamic algorithm to the later problem. Both the adaptive sampling strategy and the combinatorial algorithm could be of independent interests to the influence maximization community and the submodular maximization community. For the computaional hardness result, we exploit a novel application of the recent developed distribution PCP framework for fine grained complexity.

**Related work**   Influence maximization as a discrete optimization task is first proposed in the seminal work of Kempe et al. [35], who propose the Independent Cascade (IC) model and the Linear Threshold (LT) model, prove the submodularity property and study the performance of greedy approximation algorithm. Upon then, influence maximization and its variant has been extensively studied in the literature, including scalable algorithms [21, 13, 51, 50], adaptive algorithms [29, 28, 45, 19, 49, 6], learning algorithms [22, 8, 39, 32], etc. For detailed coverage over the area, we refer interested readers to the survey of [18, 40].

Efficient and scalable influence maximization has been the central focus of the research. Influence maximization is known to be NP-hard to approximate within $(1 - 1/e)$ under the Independent Cascade model [35], and APX-hard under the Linear Threshold model [48], while at the same, the simple greedy algorithm achieves $(1 - 1/e)$ approximation under both diffusion models, but with running time $\Omega(mnk)$. The breakthrough comes from Borgs et al. [13], who propose the Reverse influence sampling (RIS) approach and their algorithm has running time $O((m + n)k\epsilon^{-3} \log^2 n)$. The running time has been improved to $O((m + n)k\epsilon^{-2} \log n)$ and made practical in [51, 50].

There are some initial efforts on the problem of dynamic influence maximization [52, 44, 24, 3, 42]. All of them provide heuristics and none of them has rigorous theoretical guarantee on both the approximation ratio and the amortized running time. Chen et al. [24] propose Upper Bound Interchange (UBI) method with $1/2$-approximation ratio. Ohsaka et al. [44] design a RIS based algorithm that achieves $(1 - 1/e)$-approximation. Inspired by recent advance on streaming submodular maximization, Wang et al. [52] give a practical efficient algorithm that maintains a constant approximated solution. However, none of the above mentioned algorithm has rigorous guarantees on the amortized running time, and they can be as worse as $\Omega(n)$. The failure on their theoretical guarantees are not accidental, as our hardness result indicates that $\Omega(n^{1-\epsilon})$ amortized running time is essential to achieve any meaningful approximation. Meanwhile, our algorithm for incremental model substantially deviates from these approaches and provides rigorous guarantee on amortized running time.

Influence maximization is closely related to submodular maximization [15, 5, 33, 11, 10, 7, 14, 9, 25, 27, 26, 16, 25, 34, 43, 37, 23], a central area of discrete optimization. Ours is especially related to the dynamic submodular maximization problem, which has been recently studied in [43, 37, 23]. The work of [43] and [37] gives $(1/2 - \epsilon)$-approximation to the fully dynamic submodular under cardinality constraints, with amortized query complexity $O(k^2 \log^2 n \cdot \epsilon^{-3})$ and $O(\log^8 n \cdot \epsilon^{-6})$. The approximation ratio of $1/2$ is known to be tight, as Chen and Peng [23] prove that any dynamic algorithm achieves $\left(\frac{1}{2} + \epsilon\right)$ approximation must have amortized query complexity of at least $n^{\tilde{\Omega}(\epsilon)}/k^3$. The dynamic submodular maximization is studied under the query model and measured in query complexity, while we consider time complexity in dynamic influence maximization. These two are generally incomparable.

## 2 Preliminary

We consider the well-studied Independent Cascade (IC) model and the Linear Threshold (LT) model.

**Independent Cascade model**  In the IC model, the social network is described by a directed influence graph $G = (V, E, p)$, where $V$ ($|V| = n$) is the set of nodes and $E \subseteq V \times V$ ($|E| = m$) describes the set of directed edges. There is a probability $p_{u,v}$ associated with each directed edge $(u, v) \in E$. In the information diffusion process, each *activated* node $u$ has one chance to activate its out-going neighbor $v$, with independent success probability $p_{u,v}$. The *live-edge* graph $L = (V, L(E))$ is a random subgraph of $G$, where each edge $(u, v) \in E$ is included in $L(E)$ with an independent probability $p_{u,v}$. The diffusion process can also be seen as follow. At time $\tau = 0$, a live-edge graph $L$ is sampled and nodes in seed set $S \subseteq V$ are activated. At every discrete time $\tau = 1, 2, \ldots$, if a node $u$ was activated at time $\tau - 1$, then its out-going neighbors in $L$ are activated at time $\tau$. The propagation continues until no more activated nodes appears at a time step.

**Linear Threshold model**  In the LT model, the social network is a directed graph $G = (V, E, w)$, with $V$ denotes the set of nodes and $E$ denotes the set of edge. There is a weight $w_{u,v} > 0$ associate with each edge $(u, v) \in E$ and the weight satisfies $\sum_{u \in N_{\text{in}}(v)} w_{u,v} \leq 1$ for every node $v$, where $N_{\text{in}}(v)$ contains all incoming neighbors of node $v$. In the information diffusion process, a threshold $t_v$ is sampled uniformly and independently from $[0, 1]$ for each node $v$, and a node $v$ becomes active if its active in-coming neighbors have their weights exceed the threshold $t_v$. In the LT model, the live-edge graph $L = (V, L(E))$ can be obtained as follow. Each node $v$ samples an incoming neighbor from $N_{\text{in}}(v)$, where the node $u$ is sampled with probability $w_{u,v}$ and the edge $(u, v)$ is included in $L(E)$. With probability $1 - \sum_{u \in N_{\text{in}}(v)} w_{u,v}$, no edge is added. The diffusion process can also be described by the live-edge graph. That is, a live-edge graph $L$ is sampled at the beginning and the influence is spread along the graph.

**Influence maximization**  Given a seed set $S$, the influence spread of $S$, denoted as $\sigma(S)$, is the expected number of nodes activated from seed set $S$, i.e., $\sigma(S) = \mathbb{E}_{L \sim G}[|\Gamma(S, L)|]$, and $\Gamma(S, L)$ is the set of nodes reachable from $S$ in graph $L$. In the influence maximization problem, our goal is to find a seed set $S$ of size at most $k$ that maximizes the expected influence, i.e., finding $S^\star \in \arg\max_{S \subseteq V, |S| \leq k} \sigma(S)$.

The reverse reachable set [13] has been the key concept for all near-linear time IM algorithms [13, 51, 50].

**Definition 2.1** (Reverse Reachable Set)**.** *Under the IC model and the LT model, a reverse reachable (RR) set with a root node $v$, denoted $R_v$, is the random set of nodes that node $v$ reaches in one reverse propagation. Concretely, $R_v$ can be derived by randomly sampling a live-edge graph $L = (V, L(E))$ and include all nodes in $V$ that can reach $v$ under the live-edge graph $L$.*

**Dynamics of network**  Social networks are subject to changes and involve over time. We consider the following dynamic model of a social network. At each time step $t$, one of the following four types of changes could happen. (1) Introduction of a new user. This corresponds to insert a new node to influence graph; (2) Establishment of a new relationship. It is equivalent to insert a new directed edge

to the graph[2]; (3) Diminishing of an old relationship. It is equivalent to remove an existing edge of the graph; (4) Leave of an old user. This means to remove an existing node in the influence graph.

In the *incremental* model, we only allow the first two types of changes. That is, we assume the social network gets enlarged over time and we only consider the introduction of new users and new relations. In the *fully dynamic* model, we allow all four types of changes.

The theoretical results developed in this paper also adapts to other forms of change on the social network, including (a) strengthen of influence, which means the increase of the propagation probability $p_e$ for some edge under IC model or the increase of the weight $w_e$ under LT model; (b) weaken the influence. which means the decrease of propagation probability $p_e$ under IC model or the decrease of the weight $w_e$ under LT model.

**Dynamic influence maximization**    Let $G_t = (V_t, S_t)$ be the social network at time $t$. Define the expected influence to be $\sigma_t(S) = \mathbb{E}_{L \sim G_t}[|\Gamma(S, L)|]$. The goal of the dynamic influence maximization problem is to maintain a seed set $S_t$ of size at most $k$ that maximizes the expected influence at *every* time step $t$. That is to say, we aim to find $S_t^\star \in \arg\max_{S_t \subseteq V_t, |S_t| \leq k} \sigma(S_t)$ for all $t$. The (expected) amortized running time of an algorithm is defined as the (expected) average running time per update.

**Dynamic MAX-k coverage**    The (dynamic) influence maximization problem is closely related to the (dynamic) MAX-k coverage problem. In a MAX-k coverage problem, there is a collection of $n$ sets $\mathcal{A}$ defined on the ground element $[m]$. The goal is to find $k$ sets $A_1, \cdots, A_k$ such that their coverage is maximized, i.e. finding $\arg\max_{A_1,\dots,A_k \in \mathcal{A}} |\cup_{i \in [k]} A_i|$. The problem can also be formulated in terms of a bipartite graph $G = (V_L, V_R, E)$, named coverage graph, where $V_L$ ($|V_L| = n$) corresponds to set in $\mathcal{A}$ and $V_R$ ($|V_R| = m$) corresponds to element in $[m]$, for any $i \in [n], j \in [m]$, there is an edge between $V_{L,i}$ and $V_{R,j}$ if and only if $j \in A_i$, here $V_{L,i}$ is the $i$-th node in $V_L$ and $V_{R,j}$ is the $j$-th node in $V_R$. In the dynamic MAX-k coverage problem, nodes and edges arrive or leave one after another. Let $V \subseteq V_L$. At any time step $t$, we use $f_t(S)$ to denote the number of neighbors of nodes in $S$. Our goal is to maintain a set of node $S_t^* \subseteq V_L$ that maximizes the coverage for every time step $t$. That is, finding $S_t^* = \arg\max_{S_t \subseteq V_L, |S_t| \leq k} f_t(S_t)$.

**Submodular functions**    Let $V$ be a finite ground set and $f : 2^V \to \mathbb{R}$ be a set function. Given two sets $X, Y \subseteq V$, the marginal gain of $Y$ with respect to $X$ is defined as $f_X(Y) := f(X \cup Y) - f(X)$ The function $f$ is *monotone*, if for any element $e \in V$ and any set $X \subseteq V$, it holds that $f_X(e) \geq 0$. The function $f$ is *submodular*, if for any two sets $X, Y$ satisfy $X \subseteq Y \subseteq V$ and any element $e \in V \backslash Y$, one has $f_X(e) \geq f_Y(e)$. The influence spread function $\sigma$ is proved to be monotone and submodular under both the IC model and the LT model in the seminar work of [35].

## 3   Dynamic influence maximization on a growing social network

We study the dynamic influence maximization problem in the incremental model. Our main result is to show that it is possible to maintain an $(1 - 1/e - \epsilon)$-approximate solution in $k \cdot \text{poly}(\epsilon^{-1}, \log n)$ amortized time. The amortized running time is near optimal and matches the state-of-art offline algorithm up to poly-logarithmic factors.

**Theorem 3.1.** *In the incremental model, there is a randomized algorithm for the dynamic influence maximization under both Independent Cascade model and the Linear Threshold model. With probability $1 - \delta$, the algorithm maintains an $(1 - 1/e - \epsilon)$-approximately optimal solution in all time steps, with amortized running time $O(k\epsilon^{-3} \log^4(n/\delta))$.*

**Technique overview**    We provide a streamlined technique overview over our approach. All existing near-linear time IM algorithms [13, 51, 50] make use of the reverse influence sampling (RIS) technique and adopt the following two-stage framework. In the first stage, the algorithm *uniformly* samples a few RR sets, with the guarantee that the total time steps is below some threshold. In the second stage, the algorithm solves a MAX-k coverage problem on the coverage graph induced by the RR sets. Both steps are challenging to be applied in the dynamic setting and require new ideas.

---

[2]In order to adding a new directed edge $(u, v)$, one also specifies the probability $p_{u,v}$ under IC model or the weight $w_{u,v}$ under LT model.

Following the RIS framework, we first reduce the dynamic influence maximization problem to a dynamic MAX-k coverage problem (see Section 3.1). The first obstacle in dynamic setting is that it is hard to dynamically maintain an uniform distribution over nodes, and hence, one can not uniformly sample the RR sets. We circumvent it with a simple yet elegant idea, instead of uniformly sampling RR sets, we include and sample each node's RR set independently with a uniformly fixed probability $p$. This step is easy to implement in a dynamic stream, but to make the idea work, it needs some extra technique machinery. First of all, it is not obvious how to set the sampling probability $p$, as if $p$ is too large, it brings large overhead on the running time; and if it is too small, the sampled coverage graph won't yield a good approximation to the influence spread function. We use a simple trick. We first uniformly sample RR sets within some suitable amount of steps (the ESTIMATE procedure), and by doing so, we have an accurate estimate on the sampling probability (Lemma 3.3). Note that we only execute this step at beginning and it is an one-time cost. Independently sampling is friendly to dynamic algorithm, but we still need to control the total number of steps to execute it. More importantly, the sampling probability should not be a fixed one throughout the dynamic stream, since the average cost to sample a RR set could go up and down, and if the cost is too large, we need to decrease the sample probability. A simple rescue is to restart the above procedure once the coverage graph induced by independently sampling is too large, but this brings along two additional issues. First, the stopping time is highly correlated with the realization of RR sets, and this is especially problematic for independent sampling, as each incremental update on RR sets does not follow identical distribution. Second, we certainly don't want to restart the procedure for too many times. After all, if we need to restart upon every few updates, we gain nothing than the naive approach. We circumvent the first issue by maintaining two pieces of coverage graph $\mathcal{H}_{\text{est}}$ and $\mathcal{H}_{\text{cv}}$. Especially, we use $\mathcal{H}_{\text{est}}$ to estimate the average cost for sampling a RR set, and use $\mathcal{H}_{\text{cv}}$ to approximate the influence spread function. This decouples the correlation. By Martingale concentration, we can prove that (1) $\mathcal{H}_{\text{est}}$ yields good estimation on the cost of sampling RR sets (Lemma 3.5), (2) $\mathcal{H}_{\text{cv}}$ gives good approximation on the influence spread function (Lemma 3.9), and at the same time, (3) conditioning on the first event, the computation cost for $\mathcal{H}_{\text{cv}}$ won't be too large (Lemma 3.6). We circumvent the second issue by setting suitable threshold for constructing $\mathcal{H}_{\text{est}}$ (see Line 9 in INSERT-EDGE), and we periodically restart the algorithm each time the number of nodes or edges get doubled. The later step could lead to an extra $O(\log n)$ overhead, but it guarantees that each time the construction time of $\mathcal{H}_{\text{est}}$ goes above the threshold, the average steps for sampling a RR set increase by a factor of 2 (see Lemma 3.11). Hence, we restart the entire procedure no more than $O(\log n)$ times.

We next give an efficient dynamic algorithm for dynamic MAX-k coverage (Section 3.2). The offline problem can be solved in linear time via greedy algorithm, but it requires several new ideas to make the dynamic variant have low amortized cost. The major issue with offline greedy algorithm is that we don't know which element has the maximum marginal contribution until we see all elements, and in the dynamic setting, we can't afford to wait till the end. We get around it with two adaptations. First, we divide the algorithm into $O(\epsilon^{-1} \log n)$ threads[3] and for each thread, we make a guess on the optimal value (the $i$-th guess is $(1 + \epsilon)^{i-1}$). Second, instead of selecting element with the maximum margin, we maintain a threshold $(\text{OPT}_i - f(S_i))/k$ and add any element whose marginal lies above the threshold, where $S_i$ is the current solution set maintained in the $i$-th thread. We prove this still gives $(1 - 1/e)$-approximation (Lemma 3.13). The key challenge is that once we add a new element to the solution set, the threshold changes and we need to go over all existing element again. We carefully maintain a dynamic data structure and prove the amortized running time for each thread is $O(1)$ by a potential function based argument (Lemma 3.14).

### 3.1 Reducing DIM to dynamic MAX-k coverage

**Lemma 3.2.** *In the incremental model, suppose there is an algorithm that maintains an $\alpha$-approximate solution to the dynamic MAX-k coverage, with amortized running time of $\mathcal{T}_{\text{mc}}$. Then there is an algorithm, with probability at least $1 - \delta$, maintains an $(\alpha - 2\epsilon)$-approximate solution for dynamic influence maximization and has amortized running time at most $O(\mathcal{T}_{\text{mc}} k \epsilon^{-2} \log^3(n/\delta))$.*

The reduction is described in Algorithm 1-5, we provide some explanations here. From a high level, the execution of the algorithm is divided into *phases*. It restarts whenever the number of edges or the number of nodes gets doubled from the previous phase. Since the total number of steps is at most $(m + n)$, there are at most $O(\log n)$ phases. Within each phase, the algorithm is divided into

---

[3]We remark this approach has been used for submodular maximization under various setting [5, 10, 37].

*iterations*. At the beginning of each iteration, the algorithm runs the ESTIMATE procedure in order to estimate the total number of RR set it needs to sample. It then fixes a sampling probability $p$ and samples two graphs, $\mathcal{H}_{\text{cv}}$ and $\mathcal{H}_{\text{est}}$, dynamically. As explained before, $\mathcal{H}_{\text{est}}$ is used to measure the number of steps one takes for sampling RR sets, and $\mathcal{H}_{\text{cv}}$ is used to approximate the influence spread function. We stop sampling $\mathcal{H}_{\text{est}}$ and $\mathcal{H}_{\text{cv}}$ and start a new iteration whenever the number of step for constructing $\mathcal{H}_{\text{est}}$ exceeds $16Rm_0$.

Due to space limits, we defer detailed proof to the full version of this paper and outline a sketch below.

---

**Algorithm 1** INITIALIZE

1: $m \leftarrow 0, m_0 \leftarrow 0, n \leftarrow 0, n_0 \leftarrow 0$.
2: $V \leftarrow \emptyset, E \leftarrow \emptyset, \mathcal{H}_{\text{cv}} \leftarrow \emptyset, \mathcal{H}_{\text{est}} \leftarrow \emptyset$ .
3: $R \leftarrow 0, p \leftarrow 0$.

---

**Algorithm 2** ESTIMATE

1: $K \leftarrow 0$.
2: **repeat**
3:     Sample a RR set uniformly over $V$.
4:     $K \leftarrow K + 1$.
5: **until** $Rm_0$ steps have been taken
6: **return** $K$

---

**Algorithm 3** BUILDGRAPH

1: $\mathcal{H}_{\text{cv}} \leftarrow (V, \emptyset, \emptyset), \mathcal{H}_{\text{est}} \leftarrow (V, \emptyset, \emptyset)$,
2: $R \leftarrow c\epsilon^{-2}k\log(n_0/\delta)$ .
3: $K \leftarrow$ ESTIMATE().
4: $p \leftarrow \frac{K}{n_0}$.
5: **for** node $v \in V$ **do**
6:     Sample the RR set $R_{v,\text{est}}$ with prob $p$.
7:     Sample the RR set $R_{v,\text{cv}}$ with prob $p$.
8:     **for** node $u \in R_{v,\text{cv}}$ **do**
9:         INSERT-EDGE-COV($u, v$) to $\mathcal{H}_{\text{cv}}$.
10:     **end for**
11: **end for**

---

**Algorithm 4** INSERT-NODE($u$)

1: $V \leftarrow V \cup \{u\}, n \leftarrow n + 1$.
2: **if** $n \geq 2n_0$ **then**
3:     $m_0 \leftarrow m, n_0 \leftarrow n$.
4:     BUILDGRAPH().
5: **else**
6:     $V_{L,\text{est}} \leftarrow V_{L,\text{est}} \cup \{u\}$ .
7:     $V_{L,\text{cv}} \leftarrow V_{L,\text{cv}} \cup \{u\}$.
8:     $V_{R,\text{est}} \leftarrow V_{R,\text{est}} \cup \{u\}$ with prob $p$.
9:     $V_{R,\text{cv}} \leftarrow V_{R,\text{cv}} \cup \{u\}$ with prob $p$.
10: **end if**

---

**Algorithm 5** INSERT-EDGE($u, v$)

1: $E \leftarrow E \cup \{(u, v)\}, m \leftarrow m + 1$
2: **if** $m \geq 2m_0$ **then**
3:     $m_0 \leftarrow m, n_0 \leftarrow n$.
4:     BUILDGRAPH().
5: **else**
6:     **for** node $v' \in V_{R,\text{est}}$ **do**
7:         Augment the RR set $R_{v',\text{est}}$ to $R^{\text{new}}_{v',\text{est}}$.
8:         $R_{v',\text{est}} \leftarrow R^{\text{new}}_{v',\text{est}}$.
9:         **if** the total steps of building $\mathcal{H}_{\text{est}}$ exceed $16Rm_0$ **then**
10:             $m_0 \leftarrow m, n_0 \leftarrow n$.
11:             Restart and BUILDGRAPH().
12:         **end if**
13:     **end for**
14:     **for** node $v' \in V_{R,\text{cv}}$ **do**
15:         Augment the RR set $R_{v',\text{cv}}$ to $R^{\text{new}}_{v',\text{cv}}$.
16:         INSERT-EDGE-COV($u', v'$) to $\mathcal{H}_{\text{cv}}$ for all $u' \in R^{\text{new}}_{v',\text{cv}} \backslash R_{v',\text{cv}}$.
17:         $R_{v',\text{cv}} \leftarrow R^{\text{new}}_{v',\text{cv}}$.
18:     **end for**
19: **end if**

---

We first focus on one single iteration of the algorithm. Recall that in each iteration, the number of nodes satisfies $n \in [n_0, 2n_0)$ and the number of edge satisfies $m \in [m_0, 2m_0)$, and the number of steps for sampling $\mathcal{H}_{\text{est}}$ never exceeds $16Rm_0$.

**Bounds at initialization**    We prove the ESTIMATE procedure makes a good estimate on the sampling probability at the beginning of each iteration. Let $\text{AVG} \cdot m_0$ be the expected number of steps taken of sampling a uniformly random RR set, and recall $K$ is the number of RR sets ESTIMATE samples.

**Lemma 3.3.** *With probability at least* $1 - \frac{2\delta}{n^{ck/8}}$*, the number of RR sets* ESTIMATE *samples satisfies* $(1 - \epsilon) \cdot \frac{R}{\text{AVG}} \leq K \leq (1 + \epsilon) \cdot \frac{R}{\text{AVG}}$.

The initial steps to built $\mathcal{H}_{\text{est}}$ and $\mathcal{H}_{\text{cv}}$ are within $[(1 - \epsilon)^2 Rm_0, (1 + \epsilon)Rm_0]$ with high probability.

**Lemma 3.4.** *Conditioning on the event of Lemma 3.3, with probability at least $1 - \frac{4\delta}{n^{ck/8}}$, the total number of initial steps for building $\mathcal{H}_{\mathsf{est}}$ and $\mathcal{H}_{\mathsf{est}}$ are within $[(1-\epsilon)^2 R, (1+\epsilon)^2 R]$, .*

**Bounds on adaptive sampling** We dynamically sample and augment RR sets after we initiate $\mathcal{H}_{\mathsf{est}}$ and $\mathcal{H}_{\mathsf{cv}}$. Consider the $t$-th edge to arrive, let $Z_{t,i} \cdot 2m_0$ be the number of edges checked by INSERT-EDGE when it augments the RR set of the $i$-th node. Note if INSERT-NODE and BUILDGRAPH did not sample the $i$-th node and its RR set, then we simply take $Z_{t,i} = 0$. Slightly abuse of notation, we write $Z_j$ to denote the $j$-th random variable in the sequence $Z_{0,1}, \ldots, Z_{0,n_0}, Z_{1,1}, \ldots$, and we know $\{Z_j\}_{j \geq 1}$ forms a martingale. Let $J$ be the number of steps executed by INSERT-EDGE for $\mathcal{H}_{\mathsf{est}}$, our next Lemma asserts that the total number of steps is around its expectation.

**Lemma 3.5.** *Let $J_1$ be the smallest number such that $\sum_{i=1}^{J_1} \mathbb{E}[Z_i] \geq (1-\epsilon)8R$, and $J_2$ be the largest number that $\sum_{i=1}^{J_2} \mathbb{E}[Z_i] \leq (1+\epsilon)8R$. With probability at least $1 - \frac{2\delta}{n^{ck/3}}$, we have $J_1 \leq J \leq J_2$.*

With high probability, the total number of edge checked for $\mathcal{H}_{\mathsf{cv}}$ is bounded by $16(1+\epsilon)^2 Rm_0$.

**Lemma 3.6.** *Conditioning on the event of Lemma 3.5, with probability at least $1 - \frac{\delta}{n^{ck}}$, the total number of edges checked by INSERT-EDGE for $\mathcal{H}_{\mathsf{cv}}$ is at most $16(1+\epsilon)^2 Rm_0$.*

**Approximation guarantee** We prove that the coverage function induced on $\mathcal{H}_{\mathsf{cv}}$ yields a good approximation on the influence spread function. At any time step $t$, let $V_t$ be the set of node at time step $t$. For any node $v \in V_t$ and node set $S \subseteq V_t$, let $x_{t,v,S} = 1$ if $S \cap R_{v,\mathsf{cv}} \neq \emptyset$, and $x_{t,v,S} = 0$ otherwise. That is, $x_{t,v,S} = 1$ iff $S$ has non-zero intersection with the RR set of node $v$ at time step $t$. We note that if a node $v$ was not sampled by INSERT-NODE and BUILDGRAPH, and hence does not appear in $V_{R,\mathsf{cv}}$, we set $x_{t,v,S} = 0$.

**Definition 3.7** (Normalized coverage function). *Define $f_{\mathsf{cv},t} : 2^{V_t} \to \mathbb{R}^+$ to be the normalized coverage function. $f_{\mathsf{cv},t}(S) = \frac{n_0}{K} \sum_{v \in V_t} x_{t,v,S}$.*

The normalized coverage function $f_{\mathsf{cv},t}$ is an unbiased estimator on the influence spread function and it achieves good approximation with high probability.

**Lemma 3.8.** *At any time step $t$ and $S \subseteq V_t$, we have $\mathbb{E}[f_{\mathsf{cv},t}(S)] = \sigma_t(S)$, where $\sigma_t$ is the influence spread function at time $t$.*

**Lemma 3.9.** *After initializing $\mathcal{H}_{\mathsf{est}}$ and $\mathcal{H}_{\mathsf{cv}}$, at any time step $t$. Let $S \subseteq V_t$, $|S| \leq k$. Assume the condition in Lemma 3.3 holds, then we have*

$$\Pr[f_{\mathsf{cv},t}(S) > \mathbb{E}[f_{\mathsf{cv},t}(S)] + \epsilon \operatorname{OPT}_t] \leq \frac{\delta}{n^{ck/6}}, \ and, \tag{1}$$

$$\Pr[f_{\mathsf{cv},t}(S) < \mathbb{E}[f_{\mathsf{cv},t}(S)] - \epsilon \operatorname{OPT}_t] \leq \frac{\delta}{n^{ck}}. \tag{2}$$

*The expectation is taken over the randomness of the construction of $\mathcal{H}_{\mathsf{cv}}$, and $\operatorname{OPT}_t$ is defined as the maximum (expected) influence spread of a set of size $k$ at time $t$.*

The following Lemma indicates pointwise approximation carries over the approximation ratio.

**Lemma 3.10.** *Let $c \geq 2$ and let $f : 2^V \to \mathbb{R}^+$ be an arbitrary set function. Let $\mathcal{D}$ be a distribution over functions $g$ such that for all $S \subseteq V$, $\Pr_{g \sim D}[|f(S) - g(S)| - \gamma] \leq \frac{\delta}{n^{ck}}$. Let $S_g$ be an $\alpha$-approximate solution for function $g$, i.e. $g(S_g) \geq \alpha \max_{S \subseteq V, |S| \leq k} g(S)$, then we have*

$$\Pr_{g \sim D}\left[f(S_g) \leq \alpha \max_{S \subseteq V, |S| \leq k} f(S) - 2\gamma\right] \leq \frac{\delta}{n^{ck/2}}.$$

**Bounds for amortized running time** We next bound the total number of iterations within each phase. The key observation is that each time the algorithm restarts, the average steps of sampling a random RR set increases at least by a factor of 2, with high probability.

**Lemma 3.11.** *With probability at least $1 - \frac{4\delta}{n^{ck/16}}$, there are at most $O(\log n)$ iterations in a phase.*

*Proof Sketch of Lemma 3.2.* For correctness, by Lemma 3.8 and Lemma 3.9, the *normalized* coverage function $f_{\mathsf{cv},t}$ (defined on $\mathcal{H}_{\mathsf{cv}}$) guarantees $|f_{\mathsf{cv},t}(S) - \mathbb{E}[f_{\mathsf{cv},t}(S)]| = |f_{\mathsf{cv},t}(S) - \sigma_t(S)| \leq \epsilon \operatorname{OPT}_t$

**Algorithm 6** INITIALIZE

1: $\text{OPT}_i \leftarrow (1+\epsilon)^i$, $S_i = \emptyset$, $\forall i \in I$

---

**Algorithm 7** REVOKE($i$)

1: **if** there exists a node $u$ such that $f_{S_i}(u) \geq \frac{\text{OPT}_i - f(S_i)}{k}$ **and** $|S| < k$ **then**
2:     $S_i \leftarrow S_i \cup \{u\}$.
3:     REVOKE($i$).
4: **end if**

---

**Algorithm 8** INSERT-EDGE-COV($u, v$)

1: **for** $i \in I$ **do**
2:     **if** $f_{S_i}(u) \geq \frac{\text{OPT}_i - f(S_i)}{k}$ **and** $|S| < k$ **then**
3:         $S_i \leftarrow S_i \cup \{u\}$.
4:         REVOKE($i$).
5:     **end if**
6: **end for**

---

for every time step $t$. Combining with Lemma 3.10, an $\alpha$-approximate solution to the dynamic MAX-$k$ coverage problem translates to a solution set $S_t$ that satisfies $f(S) \geq (\alpha - 2\epsilon)\,\text{OPT}_t$.

For amortized running time. There are $O(\log n)$ phases, and by Lemma 3.11, there are at most $O(\log n)$ iterations in each phase. Within each iteration, the ESTIMATE procedure and the construction of $\mathcal{H}_{\text{est}}$ takes $O(m_0 ck\epsilon^{-2}\log n)$ steps in total and $O(k\epsilon^{-2}\log n)$ per update. By Lemma 3.4 and Lemma 3.6, with high probability, the construction of $\mathcal{H}_{\text{cv}}$ takes $16(1+\epsilon)^2 Rm_0$ steps in total and $O(k\epsilon^{-2}\log n)$ steps per updates. Hence, there are $O(km_0\epsilon^{-2}\log n)$ updates to the dynamic MAX-$k$ coverage problem on $\mathcal{H}_{\text{cv}}$. Taking an union bound, the overall amortized running time per update is bounded by $\log n \cdot \log n \cdot (k\epsilon^{-2}\log n + k\epsilon^{-2}\log n + k\epsilon^{-2}\log n \cdot \mathcal{T}_{\text{mc}}) \leq O(\mathcal{T}_{\text{mc}} k\epsilon^{-2}\log^3 n)$. □

### 3.2 Solving dynamic MAX-$k$ coverage in near linear time

For ease of presentation, we assume an upper bound on the value of $n$ is known and we set $I = \{0, 1, \ldots, \lceil \epsilon^{-1}\log n \rceil\}$. The algorithm maintains $|I|$ threads and for the $i$-th thread, INSERT-EDGE-COV and REVOKE augments the solution set $S_i$ only when the marginal value of a node $u$ exceeds the threshold, i.e. $f_{S_i}(u) \geq \frac{\text{OPT}_i - f(S_i)}{k}$. In particular, the threshold decreases over time and each time it decreases, REVOKE scans over all existing nodes in $V_L$. The algorithm returns the solution set with maximum value at each time step, i.e., return $\arg\max_{S_i} f_t(S_i)$.

**Theorem 3.12.** *In the incremental model, there is an algorithm for dynamic MAX-$k$ coverage that maintains a solution set with $(1 - 1/e - \epsilon)$-approximation at every time step and the amortized running time of the algorithm is at most $O(\epsilon^{-1}\log n)$.*

The approximation guarantee and the amortized running time of Algorithm 6-8 are analysed separately. We defer detailed proof to the full version of this paper and outline a sketch below.

**Lemma 3.13.** *The solution set is $(1 - 1/e - \epsilon)$-approximate to the optimal one.*

*Proof Sketch.* Let the current optimum satisfies $(1+\epsilon)^i < \text{OPT} \leq (1+\epsilon)^{i+1}$ for some $i \in I$. For the $i$-th thread, if $|S_i| < k$, then one can prove $f(S_i) \geq (1+\epsilon)^{-1}\,\text{OPT}$. On the otherside, if $|S_i| = k$. Let $s_{i,j}$ be the $j$-th element added to the set $S_i$, and denote $S_{i,j} = \{s_{i,1}, \ldots, s_{i,j}\}$. Our algorithm guarantees $f(S_{i,j+1}) - f(S_{i,j}) \geq \frac{1}{k}(\text{OPT}_i - f(S_{i,j}))$ for all $j \in [k]$. Unravel the recursion, one has $f(S_i) \geq \left(1 - \frac{1}{e} - \epsilon\right)\text{OPT}$. □

**Lemma 3.14.** *The algorithm 6-8 has amortized running time $O(\epsilon^{-1}\log k)$.*

*Proof Sketch.* We prove that for each thread $i \in I$, the amortized running time is $O(1)$. Let $V_R^i \subseteq V_R$ be nodes covered by $S_i$. Let $P(t)$ denote the number of operations performed on the $i$-th thread up to time $t$. For each edge $e = (u, v)$, let $X_e$ denote whether the edge $e$ is covered by $V_R^i$, i.e. $X_e = 1$ if $v \in V_R^i$ and $X_e = 0$ otherwise. Similarly, for each node $v \in V_R$, let $Y_v$ denote whether node $v$ is included in $V_R^i$, i.e., $Y_v = 1$ if $v \in V_R^i$ and $Y_v = 0$ otherwise. For each node $u \in V_L$, let $Z_u$ denote whether node $u$ is included in $S_i$, i.e., $Y_v = 1$ if $v \in S_i$ and $Y_v = 0$ otherwise. Define the potential function $\Phi : t \to \mathbb{R}^+$:

$$\Phi(t) := 2|V_t| + 2|E_t| + 2\sum_{e \in E_t} X_e + \sum_{v \in V_R} Y_v + 2\sum_{v \in V_L} Z_u.$$

We prove $P(t) \leq \Phi(t)$ always holds. This suffices for our purpose as one can easily show $\Phi(t) \leq 5t$. The claim is executed by an induction showing $P(t) - P(t-1) \leq \Phi(t) - \Phi(t-1)$ holds for $t$. □

Taking $\alpha = 1 - 1/e$ and $\mathcal{T}_{\mathsf{mc}} = O(\epsilon^{-1} \log n)$, we wrap up the proof of Theorem 3.1.

## 4 Fully dynamic influence maximization

In the fully dynamic model, the social network involves over time and all four types of change exhibit. Our main delivery is a sharp computational hardness result. We prove that under the SETH, unless the amortized running time is $n^{1-o(1)}$, the approximation ratio can not even be $2^{-(\log n)^{1-o(1)}}$. We first provide some background on fine-grain complexity. We reduce from the Strong Exponential Time Hypothesis (SETH), which is a pessimistic version of $P \neq NP$ that postulates that much better algorithms for $k$-SAT do not exist. The SETH is first proposed in the seminal work of [31] and it is widely believed in the computational complexity, see [53, 47] for detailed survey.

**Conjecture 4.1** (Strong Exponential Time Hypothesis (SETH), [31])**.** *For any $\epsilon > 0$, there exists $k \geq 3$ such that $k$-SAT on variables can not be solved in time $O(2^{(1-\epsilon)n})$.*

Our starting point is the following SETH-based hardness of approximation result. The result is proven in [2, 17] using the distributed PCP framework [2, 46] for hardness of approximation results in P.

**Theorem 4.2** ([2, 17, 1])**.** *Let $\epsilon > 0$, $m = n^{o(1)}$ and $t = 2^{(\log n)^{1-o(1)}}$. Given two collections of $n$ sets $\mathcal{A}$ and $\mathcal{B}$ over universe $[m]$. Unless SETH is false, no algorithm can distinguish the following two cases in $O(n^{2-\epsilon})$ :*

**YES instance.** *There exists two sets $A \in \mathcal{A}$, $B \in \mathcal{B}$ such that $B \subseteq A$;*

**NO instance.** *For every $A \in \mathcal{A}$, $B \in \mathcal{B}$ we have $|A \cap B| < |B|/t$.*

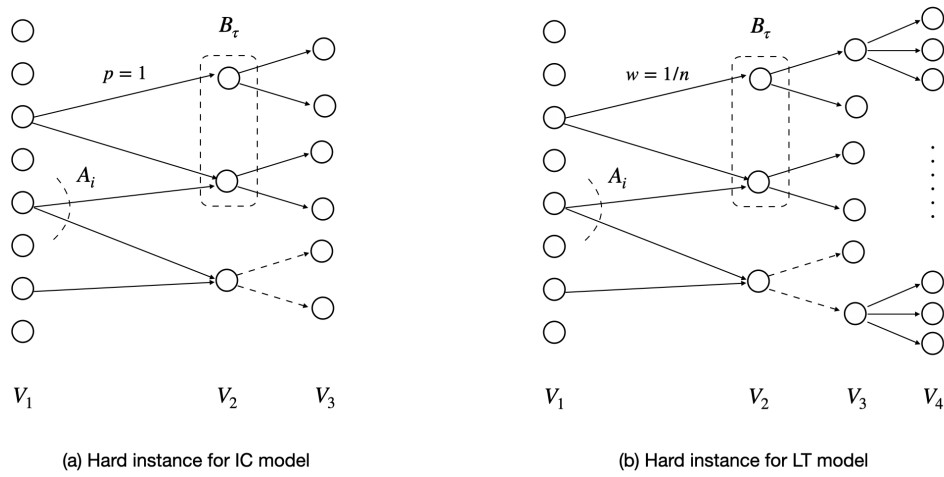

(a) Hard instance for IC model          (b) Hard instance for LT model

Figure 1: The hard instance for fully dynamic influence maximization.

**Theorem 4.3.** *Assuming SETH, in the fully dynamic influence maximization problem under IC model, no algorithm can achieve $2^{-(\log n)^{1-o(1)}}$ approximation unless the amortized running time is $n^{1-o(1)}$.*

*Proof Sketch.* Let $m = n^{o(1)}$, $t = 2^{(\log n)^{1-o(1)}}$, $k = 1$. Given an instance $\mathcal{A}$, $\mathcal{B}$ of the problem in Theorem 4.2, we reduce it to DIM. We assume $|B_\tau| \geq t$ for all $\tau \in [n]$ as we can duplicate the ground element for $t$ times. Consider the following influence graph $G = (V, E, p)$, where the influence probability on all edges are 1. The node set $V$ is partitioned into $V = V_1 \cup V_2 \cup V_3$, $|V_1| = n, |V_2| = m, |V_3| = m^2 t$. Intuitively, the $i$-th node $v_{1,i}$ of $V_1$ corresponds to the set $A_i \in \mathcal{A}$, and the $j$-th node $v_{2,j}$ in $V_1$ correspond to the $j$-th element of ground set $[m]$. There is a directed

edge from $v_{1,i}$ to $v_{2,j}$, iff the $j$-th element is contained in the set $A_1$. We write nodes in $V_3$ as $\{V_{3,j,\ell}\}_{j\in[m],\ell\in[mt]}$, and there is a directed edge node $v_{2,j}$ to $v_{3,j,\ell}$, $j \in [m], \ell \in [mt]$.

Consider the following update sequence of the DIM problem. The graph $G$ is loaded first and then followed by $n$ consecutive epochs. In the $\tau$-th epoch, all edges between $V_2$ and $V_3$ are deleted, and for each $j \in B_\tau$, we add back the edge between $v_{2,j}$ and $v_{3,j,\ell}$ for all $\ell \in [mt]$.

One can show the total number of updates is at most $n^{1+o(1)}$. We prove by contrary and suppose there is an algorithm for DIM that achieves $2/t$-approximation in $n^{1-\epsilon}$ time. Under the above reduction, we output YES, if for some epoch $\tau \in [n]$, the DIM algorithm outputs a solution with influence spread greater than $2m|B_\tau|$. We output NO otherwise. One can prove (1) if there exists $A_i \in \mathcal{A}$, $B_\tau \in \mathcal{B}$ such that $B_\tau \subseteq A_i$, then we output YES, and, (2) if $|A_i \cap B_\tau| < |B_\tau|/t$ for any $i, \tau \in [n]$, the algorithm outputx NO. Hence, we conclude under SETH, there is no $2/t$-approximation algorithm unless the amortized running time is $O(n^{1-\epsilon})$. $\qquad\square$

The lower bound can be extended to the LT model, under the additional constraints that the algorithm only selects seeds from a prescribed set $V' \subseteq V$. This a natural assumption that has been made/discussed in previous work [20, 36, 48] for the LT model. The construction is similar to Theorem 4.4, with the exception that (1) the weight on edges between $V_1$ and $V_2$ are $1/n$ and all other edges have weight 1, (2) the node set $V$ is partitioned into four parts $V_1 \cup V_2 \cup V_3 \cup V_4$. $|V_4| = nm^2t$ and each node in $V_3$ is connected to $n$ nodes in $V_4$. Detailed proof can be found in the full version of this paper.

**Theorem 4.4.** *Assuming SETH, for the fully dynamic influence maximization problem under LT model, if the algorithm is only allowed to select seed from a prescribed set, then no algorithm can achieve $2^{-(\log n)^{1-o(1)}}$ approximation unless the amortized running time is $n^{1-o(1)}$.*

## 5 Discussion

We study the dynamic influence maximization problem and provide sharp computational results on the incremental update model and the fully dynamic model. In the incremental model, we provide an algorithm that maintains a seed set with $(1 - 1/e - \epsilon)$-approximation and has amortized running time $k \cdot \text{poly}(\log n, \epsilon^{-1})$, which matches the state of art offline IM algorithm up to poly-logarithmic factor. For the fully dynamic model, we prove that under SETH, no algorithm can achieve $2^{-(\log n)^{1-o(1)}}$ approximation unless the amortized running time is $n^{1-o(1)}$. There are a few interesting questions for future investigation: (1) Further improve the amortized running time in the incremental model. In particular, is it possible to reduce the amortized running time of the dynamic MAX-k coverage procedure to $O(1)$? (2) Investigate fully dynamic influence maximization problem under mild assumptions, e.g. what if the graph is bipartite?

## Acknowledgement

Binghui Peng wishes to thank Xi Chen for useful discussions on dynamic submodular maximization, and thank Matthew Fahrbach for useful comments. Binghui Peng is supported in part by Christos Papadimitriou's NSF grants CCF-1763970 AF, CCF-1910700 AF and a softbank grant, and by Xi Chen's NSF grants NSF CCF-1703925.

## Broader Impact

Our work is mainly theoretical with no foreseeable ethical issues.

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
