8:         $R_{v',\text{est}} \leftarrow R_{v',\text{est}}^{\text{new}}$.
9:         **if** the total steps of building $\mathcal{H}_{est}$ exceed $16Rm_0$ **then**
10:             $m_0 \leftarrow m, n_0 \leftarrow n$.
11:             Restart and BUILDGRAPH().
12:         **end if**
13:     **end for**
14:     **for** node $v' \in V_{R,\text{cv}}$ **do**
15:         Augment the RR set $R_{v',\text{cv}}$ to $R_{v',\text{cv}}^{\text{new}}$.
16:         INSERT-EDGE-COV$(u', v')$ to $\mathcal{H}_{cv}$ for all $u' \in R_{v',\text{cv}}^{\text{new}} \setminus R_{v',\text{cv}}$.
17:         $R_{v',\text{cv}} \leftarrow R_{v',\text{cv}}^{\text{new}}$.
18:     **end for**
19: **end if**

---

We first focus on one single iteration of the algorithm. Recall that in each iteration, the number of nodes satisfies $n \in [n_0, 2n_0)$ and the number of edge satisfies $m \in [m_0, 2m_0)$, and the number of steps for sampling $\mathcal{H}_{est}$ never exceeds $16Rm_0$.

**Bounds at initialization** We prove the ESTIMATE procedure makes a good estimate on the sampling probability at the beginning of each iteration. Let $\text{AVG} \cdot m_0$ be the expected number of steps taken of sampling a uniformly random RR set, and recall $K$ is the number of RR sets ESTIMATE samples.

**Lemma 3.3.** *With probability at least* $1 - \frac{2\delta}{n^{ck/8}}$, *the number of RR sets* ESTIMATE *samples satisfies* $(1 - \epsilon) \cdot \frac{R}{\text{AVG}} \leq K \leq (1 + \epsilon) \cdot \frac{R}{\text{AVG}}$.

The initial steps to built $\mathcal{H}_{est}$ and $\mathcal{H}_{cv}$ are within $[(1 - \epsilon)^2 Rm_0, (1 + \epsilon)Rm_0]$ with high probability.

**Lemma 3.4.** *Conditioning on the event of Lemma 3.3, with probability at least $1 - \frac{4\delta}{n^{ck/8}}$, the total number of initial steps for building $\mathcal{H}_{\text{est}}$ and $\mathcal{H}_{\text{est}}$ are within $[(1-\epsilon)^2 R, (1+\epsilon)^2 R], .$*

**Bounds on adaptive sampling** We dynamically sample and augment RR sets after we initiate $\mathcal{H}_{\text{est}}$ and $\mathcal{H}_{\text{cv}}$. Consider the $t$-th edge to arrive, let $Z_{t,i} \cdot 2m_0$ be the number of edges checked by INSERT-EDGE when it augments the RR set of the $i$-th node. Note if INSERT-NODE and BUILDGRAPH did not sample the $i$-th node and its RR set, then we simply take $Z_{t,i} = 0$. Slightly abuse of notation, we write $Z_j$ to denote the $j$-th random variable in the sequence $Z_{0,1}, \ldots, Z_{0,n_0}, Z_{1,1}, \ldots$, and we know $\{Z_j\}_{j \geq 1}$ forms a martingale. Let $J$ be the number of steps executed by INSERT-EDGE for $\mathcal{H}_{\text{est}}$, our next Lemma asserts that the total number of steps is around its expectation.

**Lemma 3.5.** *Let $J_1$ be the smallest number such that $\sum_{i=1}^{J_1} \mathbb{E}[Z_i] \geq (1-\epsilon)8R$, and $J_2$ be the largest number that $\sum_{i=1}^{J_2} \mathbb{E}[Z_i] \leq (1+\epsilon)8R$. With probability at least $1 - \frac{2\delta}{n^{ck/3}}$, we have $J_1 \leq J \leq J_2$.*

With high probability, the total number of edge checked for $\mathcal{H}_{\text{cv}}$ is bounded by $16(1+\epsilon)^2 Rm_0$.

**Lemma 3.6.** *Conditioning on the event of Lemma 3.5, with probability at least $1 - \frac{\delta}{n^{ck}}$, the total number of edges checked by INSERT-EDGE for $\mathcal{H}_{\text{cv}}$ is at most $16(1+\epsilon)^2 Rm_0$.*

**Approximation guarantee** We prove that the coverage function induced on $\mathcal{H}_{\text{cv}}$ yields a good approximation on the influence spread function. At any time step $t$, let $V_t$ be the set of node at time step $t$. For any node $v \in V_t$ and node set $S \subseteq V_t$, let $x_{t,v,S} = 1$ if $S \cap R_{v,\text{cv}} \neq \emptyset$, and $x_{t,v,S} = 0$ otherwise. That is, $x_{t,v,S} = 1$ iff $S$ has non-zero intersection with the RR set of node $v$ at time step $t$. We note that if a node $v$ was not sampled by INSERT-NODE and BUILDGRAPH, and hence does not appear in $V_{R,\text{cv}}$, we set $x_{t,v,S} = 0$.

**Definition 3.7** (Normalized coverage function). *Define $f_{\text{cv},t} : 2^{V_t} \to \mathbb{R}^+$ to be the normalized coverage function. $f_{\text{cv},t}(S) = \frac{n_0}{K} \sum_{v \in V_t} x_{t,v,S}$.*

The normalized coverage function $f_{\text{cv},t}$ is an unbiased estimator on the influence spread function and it achieves good approximation with high probability.

**Lemma 3.8.** *At any time step $t$ and $S \subseteq V_t$, we have $\mathbb{E}[f_{\text{cv},t}(S)] = \sigma_t(S)$, where $\sigma_t$ is the influence spread function at time $t$.*

**Lemma 3.9.** *After initializing $\mathcal{H}_{\text{est}}$ and $\mathcal{H}_{\text{cv}}$, at any time step $t$. Let $S \subseteq V_t$, $|S| \leq k$. Assume the condition in Lemma 3.3 holds, then we have*

$$\Pr\left[f_{\text{cv},t}(S) > \mathbb{E}[f_{\text{cv},t}(S)] + \epsilon \operatorname{OPT}_t\right] \leq \frac{\delta}{n^{ck/6}}, \ and, \tag{1}$$

$$\Pr\left[f_{\text{cv},t}(S) < \mathbb{E}[f_{\text{cv},t}(S)] - \epsilon \operatorname{OPT}_t\right] \leq \frac{\delta}{n^{ck}}. \tag{2}$$

*The expectation is taken over the randomness of the construction of $\mathcal{H}_{\text{cv}}$, and $\operatorname{OPT}_t$ is defined as the maximum (expected) influence spread of a set of size $k$ at time $t$.*

The following Lemma indicates pointwise approximation carries over the approximation ratio.

**Lemma 3.10.** *Let $c \geq 2$ and let $f : 2^V \to \mathbb{R}^+$ be an arbitrary set function. Let $\mathcal{D}$ be a distribution over functions $g$ such that for all $S \subseteq V$, $\Pr_{g \sim D}[|f(S) - g(S)| - \gamma] \leq \frac{\delta}{n^{ck}}$. Let $S_g$ be an $\alpha$-approximate solution for function $g$, i.e. $g(S_g) \geq \alpha \max_{S \subseteq V, |S| \leq k} g(S)$, then we have*

$$\Pr_{g \sim D}\left[f(S_g) \leq \alpha \max_{S \subseteq V, |S| \leq k} f(S) - 2\gamma\right] \leq \frac{\delta}{n^{ck/2}}.$$

**Bounds for amortized running time** We next bound the total number of iterations within each phase. The key observation is that each time the algorithm restarts, the average steps of sampling a random RR set increases at least by a factor of 2, with high probability.

**Lemma 3.11.** *With probability at least $1 - \frac{4\delta}{n^{ck/16}}$, there are at most $O(\log n)$ iterations in a phase.*

*Proof Sketch of Lemma 3.2.* For correctness, by Lemma 3.8 and Lemma 3.9, the *normalized* coverage function $f_{\text{cv},t}$ (defined on $\mathcal{H}_{\text{cv}}$) guarantees $|f_{\text{cv},t}(S) - \mathbb{E}[f_{\text{cv},t}(S)]| = |f_{\text{

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

## A  Probabilistic tools

**Lemma A.1** (Chernoff bound, the multiplicative form). *Let $X = \sum_{i=1}^{n} X_i$, where $X_i \in [0,1]$ are independent random variables. Let $\mu = \mathbb{E}[X] = \sum_{i=1}^{n} \mathbb{E}[X_i]$. Then*
*1. $\Pr[X \geq (1+\delta)\mu] \leq \exp(-\delta^2\mu/(2+\delta))$, $\forall \delta > 0$ ;*
*2. $\Pr[X \leq (1-\delta)\mu] \leq \exp(-\delta^2\mu/2)$, $\forall 0 < \delta < 1$.*

**Lemma A.2** (Hoeffding bound). *Let $X_1, \cdots, X_n$ denote $n$ independent bounded variables in $[a_i, b_i]$. Let $X = \sum_{i=1}^{n} X_i$, then we have*

$$\Pr[|X - \mathbb{E}[X]| \geq t] \leq 2\exp\left(-\frac{2t^2}{\sum_{i=1}^{n}(b_i - a_i)^2}\right).$$

**Lemma A.3** (Azuma bound, the multiplicative form). *Let $X_1, \cdots, X_n \in [0,1]$ be real valued random variable. Suppose*
$$\mathbb{E}[X_i | X_1, \cdots, X_n] = \mu_i$$
*holds for all $i \in [n]$ and let $\mu = \sum_{i=1}^{n} \mu_i$. Then, we have*

$$\Pr[\sum_{i=1}^{n} X_i \geq (1+\delta)\mu] \leq \exp\left(-\delta^2\mu/(2+\delta)\right)$$

$$\Pr[\sum_{i=1}^{n} X_i \leq (1-\delta)\mu] \leq \exp\left(-\delta^2\mu/2\right).$$

**Lemma A.4** (Azuma-Hoeffding bound). *Let $X_0, \cdots, X_n$ be a martingale sequence with respect to the filter $F_0 \subseteq F_2 \cdots \subseteq F_n$ such that for $Y_i = X_i - X_{i-1}$, $i \in [n]$, we have that $|Y_i| = |X_i - X_{i-1}| \leq c_i$. Then*

$$\Pr[|X_t - Y_0| \geq t] \leq 2\exp\left(-\frac{t^2}{2\sum_{i=1}^{n} c_i^2}\right).$$

## B  Missing proof from Section 3.1

We first provide a proof of Lemma 3.3, which states the ESTIMATE procedure gives a good estimate on sampling probability.

*Proof of Lemma 3.3.* Let $X_t$ be the fraction of edges that are checked in the $t$-th iteration of ESTI-MATE (i.e., Line 3 of ESTIMATE). By definition, we have that $\mathbb{E}[X_t] = \text{AVG}$ and $X_t \in [0,1]$ for all $t$. Note that $K$ is precisely the minimum value $K'$ such that $\sum_{t=1}^{K'} X_1 \geq R$.

Let $K_1 = (1-\epsilon) \cdot \frac{R}{\text{AVG}}$ and $K_2 = (1+\epsilon) \cdot \frac{R}{\text{AVG}}$. We want to show that with high probability $K \in [K_1, K_2]$, Note that this event is exactly the intersection of the event $\sum_{t=1}^{K_1} X_t < R$ and the $\sum_{t=1}^{K_2} X_t > R$. For the first one, notice that

$$\mathbb{E}\left[\sum_{t=1}^{K_1} X_t\right] = K_1 \cdot \text{AVG} = (1-\epsilon)R.$$

By the multiplicative form Chernoff bound, we have

$$\Pr\left[\sum_{t=1}^{K_1} X_t \geq R\right] \leq \Pr\left[\sum_{t=1}^{K_1} X_t \geq (1-\epsilon)\sum_{t=1}^{K_1} \mathbb{E}[X_t]\right] \leq \exp\left(-\epsilon^2 \sum_{t=1}^{K_1} \mathbb{E}[X_t]/3\right)$$
$$\leq \exp(-\epsilon^2(1-\epsilon)ck\epsilon^{-2}\log(n/\delta)/3) \leq \delta/n^{ck/6}.$$

For the second event, notice that

$$\mathbb{E}\left[\sum_{t=1}^{K_2} X_t\right] = K_2 \cdot \text{AVG} = (1+\epsilon)R,$$

and by the multiplicative form Chernoff bound, we have

$$\Pr\left[\sum_{t=1}^{K_2} X_t \le R\right] < \Pr\left[\sum_{t=1}^{K_2} X_t \le (1-\epsilon/2)\sum_{t=1}^{K_2}\mathbb{E}[X_t]\right] \le \exp\left(-\epsilon^2\sum_{t=1}^{K_2}\mathbb{E}[X_t]/8\right)$$

$$\le \exp(-\epsilon^2(1+\epsilon)ck\epsilon^{-2}\log(n/\delta)/8) \le \delta/n^{ck/8}.$$

Taking an union bound, we conclude that with probability at least $1 - \frac{2\delta}{n^{ck/8}}$, $K \in [K_1, K_2]$. $\qquad\square$

We next prove Lemma 3.4, which states the initial steps for building $\mathcal{H}_{\text{est}}$ and $\mathcal{H}_{\text{cv}}$ is not large.

*Proof of Lemma 3.4.* We prove that with $1 - \frac{2\delta}{n^{ck/8}}$, the total number of steps for building $\mathcal{H}_{\text{est}}$ are within $[(1-\epsilon)^2 R, (1+\epsilon)^2 R]$, the Lemma then follows by an union bound. Let $Y_i$ ($i \in [n_0]$) be the fraction of edges checked when sampling the RR set $R_i$ of the $i$-th node. Notice that if with probability $1 - p$, the algorithm won't sample $R_i$, and we take $Y_i = 0$ for this case. We have that $Y_i \in [0, 1]$ and

$$\mathbb{E}\left[\sum_{i=1}^{n_0} Y_t\right] = \frac{K}{n_0} \cdot n_0 \, \text{AVG} = K \cdot \text{AVG} \in [(1-\epsilon)R, (1+\epsilon)R]. \tag{3}$$

The last step holds due to Lemma 3.3.

Consequently, by the multiplicative form Chernoff bound, we have

$$\Pr\left[\sum_{i=1}^{n_0} Y_i \ge (1+\epsilon)^2 R\right] \le \Pr\left[\sum_{i=1}^{n_0} Y_i \ge (1+\epsilon)\sum_{i=1}^{n_0}\mathbb{E}[Y_i]\right] \le \exp\left(-\epsilon^2\sum_{i=1}^{n_0}\mathbb{E}[Y_i]/3\right)$$

$$\le e^{-\epsilon^2(1-\epsilon)ce^{-2}k\log(n/\delta)/3} \le \delta/n^{ck/6}.$$

The first step and third follows from Eq. (3). We assume $\epsilon < 1/2$ in the last step.

Similarly, one has

$$\Pr\left[\sum_{i=1}^{n_0} Y_i \le (1-\epsilon)^2 R\right] \le \Pr\left[\sum_{i=1}^{n_0} Y_i \le (1-\epsilon)\sum_{i=1}^{n_0}\mathbb{E}[Y_i]\right] \le \exp\left(-\epsilon^2\sum_{i=1}^{n_0}E[Y_i]/2\right)$$

$$\le e^{-\epsilon^2(1-\epsilon)ce^{-2}k\log(n/\delta)/2} \le \delta/n^{ck/8}.$$

We conclude the proof here. $\qquad\square$

We next provide a proof for Lemma 3.5, which states that the total number of steps of constructing $\mathcal{H}_{\text{est}}$ is around its expectation.

*Proof of Lemma 3.5.* We know that the random variable $Z_j \in [0, 1]$ since the total number edge checked for one RR set is at most $m \le 2m_0$. Moreover, $\{Z_j\}_{j\ge1}$ forms a martingale. Since $J$ is the first number such that $\sum_{j=1}^{J} 2m_0 Z_j \ge 16Rm_0$, i.e, $\sum_{j=1}^{J} Z_j \ge 8Rm_0$, the event $J \in [J_1, J_2]$ is equivalent to the intersection of event $\sum_{j=1}^{J_1} Z_j > 8R$ and $\sum_{j=1}^{J_2} Z_j < 8R$. We bound the probability separately.

For the first event, we have

$$(1-\epsilon)8R \le \sum_{j=1}^{J_1}\mathbb{E}[Z_j] \le (1-\epsilon)8R + 1.$$

By the multiplicative form Azuma bound, we have

$$\Pr\left[\sum_{j=1}^{J_1} Z_j > 8R\right] \le \Pr\left[\sum_{j=1}^{J_1} Z_j \ge (1+\epsilon/2)\sum_{j=1}^{J_1}\mathbb{E}[Z_j]\right] \le \exp\left(-\epsilon^2\sum_{j=1}^{J_1}\mathbb{E}[Z_j]/12\right)$$

$$\le \exp(-\epsilon^2(1-\epsilon)8ce^{-2}k\log(n/\delta)/12) \le \frac{\delta}{n^{ck/3}}. \tag{4}$$

Similarly, for the second event, we have that

$$(1+\epsilon)8R - 1 \le \sum_{j=1}^{J_2} \mathbb{E}[Z_j] \le (1+\epsilon)8R.$$

By the multiplicative form Azuma bound, we have

$$\Pr\left[\sum_{j=1}^{J_2} Z_j < 8R\right] \le \Pr\left[\sum_{j=1}^{J_2} Z_j \le (1-\epsilon/2)\sum_{j=1}^{J_2} \mathbb{E}[Z_j]\right] \le \exp\left(-\epsilon^2 \sum_{j=1}^{J_2} \mathbb{E}[Z_j]/8\right)$$

$$\le \exp(-\epsilon^2 8 c\epsilon^{-2} k \log(n/\delta)/8) \le \frac{\delta}{n^{ck}} \tag{5}$$

Combine the Eq. (4) and Eq. (5) with union bound, we conclude that probability at least $1 - \frac{2\delta}{n^{ck/3}}$, the total number step for $\mathcal{H}_{\text{est}}$ satisfies $J_1 \le J \le J_2$. $\qquad \square$

The total number of edge checked for $\mathcal{H}_{\text{cv}}$ is bounded by $16(1+\epsilon)^2 R m_0$ with high probability, as stated in Lemma 3.6

*Proof of Lemma 3.6.* Similar as Lemma 3.5, consider the $t$-th edge to arrive, let $W_{t,i} \cdot 2m_0$ be the number of edges checked by INSERT-EDGE when it augments the RR set of the $i$-th node on $\mathcal{H}_{\text{cv}}$. Again, if INSERT-NODE and BUILDGRAPH did not sample the $i$-th node and its RR set, we simply have $W_{t,i} = 0$. We also write $W_j$ to denote the $j$-th random variable in the sequence $W_{0,1}, \ldots, W_{0,n_0}, W_{1,1}, \ldots$, and it also forms a martingale. Recall $J$ is the total number of steps executed by INSERT EDGE on constructing $\mathcal{H}_{\text{est}}$, and it is also the time where we stop augmenting $\mathcal{H}_{\text{cv}}$.

We know $W_j \in [0,1]$ since $m \le 2m_0$ and $\{W_j\}_{j\ge 1}$ forms a martingale. Since we sample $\mathcal{H}_{\text{est}}$ and $\mathcal{H}_{\text{cv}}$ separately, the stopping time is irrelevant to the realization of $W_j$. Condition on the event of Lemma 3.5, i.e. $J \in [J_1, J_2]$, we know that

$$(1-\epsilon)8R \le \sum_{j=1}^{J_1} \mathbb{E}[W_j] \le \sum_{j=1}^{J} \mathbb{E}[W_j] \le \sum_{j=1}^{J_2} \mathbb{E}[W_j] \le (1+\epsilon)8R.$$

Hence, by Azuma bound, we have

$$\Pr\left[\sum_{j=1}^{J} W_j \cdot 2m_0 \ge 16(1+\epsilon)^2 R m_0\right] = \Pr\left[\sum_{j=1}^{J} W_j \ge (1+\epsilon)^2 8R\right]$$

$$= \Pr\left[\sum_{i=1}^{J} W_j \ge (1+\epsilon)\sum_{i=1}^{J} \mathbb{E}[Z_j]\right]$$

$$\le \exp\left(-\epsilon^2 \sum_{j=1}^{J} \mathbb{E}[W_j]/3\right)$$

$$\le \exp(-\epsilon^2 8(1-\epsilon)c\epsilon^{-2} k \log(\delta/n)/3)$$

$$\le \frac{\delta}{n^{ck}}.$$

This concludes the proof. $\qquad \square$

We provide the proof of Lemma 3.8, which asserts that the normalized coverage function $f_{\text{cv},t}$ is an unbiased estimator on the influence spread function.

*Proof of Lemma 3.8.* The expected influence spread of a set $S$ at time step $t$ satisfies

$$\sigma_t(S) = \sum_{v \in V_t} \mathbb{E}[|S \cap R_{v,t}|] = \sum_{v \in V_t} \frac{n_0}{K} \mathbb{E}[x_{t,v,S}] = \mathbb{E}[f_{\text{cv},t}(S)].$$

The first step follows from the definition of influence spread function $\sigma_t$, the second step follows from our sampling process, the definition of $x_{t,v,S}$ and the fact that we sample the RR set of node $v$ with probability $p = \frac{K}{n_0}$. $\qquad\square$

Next, we prove Lemma 3.9.

*Proof of Lemma 3.9.* Fix a fixed time step $t$. For any node set $S \subseteq V_t$, $|S| \leq k$, by Lemma 3.8, we have $\mathbb{E}[f_{\mathsf{cv},t}(S)] = \sigma_t(S) \leq \mathrm{OPT}_t$. For convenience, we write $\mathbb{E}[f_{\mathsf{cv},t}(S)] = \lambda\,\mathrm{OPT}_t$, and $\lambda \in [0,1]$. Hence, we have

$$\Pr\left[f_{\mathsf{cv},t}(S) > \mathbb{E}[f_{\mathsf{cv},t}(S)] + \epsilon\,\mathrm{OPT}_t\right] = \Pr\left[f_{\mathsf{cv},t}(S) > (1 + \frac{\epsilon}{\lambda})\mathbb{E}[f_{\mathsf{cv},t}(S)]\right]$$

$$= \Pr\left[\sum_{v \in V_t} x_{t,v,S} > (1 + \frac{\epsilon}{\lambda})\sum_{v \in V_t}\mathbb{E}[x_{t,v,S}]\right]. \quad (6)$$

We divide into two cases. First, suppose $\lambda > \epsilon$. Since $x_{t,v,S} \in \{0,1\}$ and they are indepedent, by the multiplicative form Chernoff bound, we have

$$\Pr\left[\sum_{v \in V_t} x_{t,v,S} > (1 + \frac{\epsilon}{\lambda})\sum_{v \in V_t}\mathbb{E}[x_{t,v,S}]\right] \leq \exp\left(-\epsilon^2\,\mathbb{E}[\sum_{v \in V_t} x_{t,v,S}]/3\lambda^2\right).$$

The exponent obeys

$$\epsilon^2 \sum_{v \in V_t}\mathbb{E}[x_{t,v,S}]/3\lambda^2 = \epsilon^2\frac{K}{n_0}\lambda\,\mathrm{OPT}_t\,/3\lambda^2 \geq \epsilon^2\frac{K}{3n_0}\mathrm{OPT}_t \geq \epsilon^2(1-\epsilon)\frac{R}{3n_0\,\mathrm{AVG}}\mathrm{OPT}_t$$

$$\geq \epsilon^2(1-\epsilon)R/3 \geq ck\log(n/\delta)/6. \quad (7)$$

where the first step follows from

$$\frac{n_0}{K}\sum_{v \in V_t}\mathbb{E}[x_{t,v,S}] = \mathbb{E}[f_{\mathsf{cv},t}(S)] = \lambda\,\mathrm{OPT}_t, \quad (8)$$

the second step follows from $\lambda < 1$, the third comes from the condition of Lemma 3.3, i.e., $K \in [(1-\epsilon)\cdot\frac{R}{\mathrm{AVG}}, (1+\epsilon)\cdot\frac{R}{\mathrm{AVG}}]$. The fourth step follows from Lemma B.1 and the monotonicity of OPT, i.e.,

$$\mathrm{AVG} \leq \frac{\mathrm{OPT}_0}{n_0} \leq \frac{\mathrm{OPT}_t}{n_0}. \quad (9)$$

Hence, when $\epsilon < \lambda$, one has

$$\Pr\left[f_{\mathsf{cv},t}(S) > \mathbb{E}[f_{\mathsf{cv},t}(S)] + \epsilon\,\mathrm{OPT}_t\right] \leq \frac{\delta}{n^{ck/6}}.$$

Next, suppose $\lambda \leq \epsilon$, then $\epsilon/\lambda \geq 1$. The multiplicative form Chernoff bound

$$\Pr\left[\sum_{v \in V_t} x_{t,v,S} > (1 + \frac{\epsilon}{\lambda})\sum_{v \in V_t}\mathbb{E}[x_{t,v,S}]]\right] \leq \exp\left(-\epsilon\sum_{v \in V_t}\mathbb{E}[x_{t,v,S}]/3\lambda\right)$$

The exponent satisfies

$$\epsilon\sum_{v \in V_t}\mathbb{E}[x_{t,v,S}]/3\lambda = \epsilon\cdot\frac{K}{3n_0}\mathrm{OPT}_t \geq \epsilon(1-\epsilon)\frac{R}{3n_0\,\mathrm{AVG}}\mathrm{OPT}_t \geq \epsilon(1-\epsilon)R/3 \geq ck\log(n/\delta)/6.$$

where the first step follows from Eq. (8), the second step follows from the condition of Lemma 3.3, i.e., $K \in [(1-\epsilon)\cdot\frac{R}{\mathrm{AVG}}, (1+\epsilon)\cdot\frac{R}{\mathrm{AVG}}]$. The third step follows Eq. (9). Hence, when $\lambda \geq \epsilon$, one also has

$$\Pr\left[f_{\mathsf{cv},t}(S) > \mathbb{E}[f_{\mathsf{cv},t}(S)] + \epsilon\,\mathrm{OPT}_t\right] \leq \frac{\delta}{n^{ck/6}}.$$

This proves the Eq. (1). The proof of Eq. (2) is similar. In particular, we have

$$\Pr\left[f_{\mathsf{cv},t}(S) < \mathbb{E}[f_{\mathsf{cv},t}(S)] - \epsilon\,\mathrm{OPT}_t\right] = \Pr\left[f_{\mathsf{cv},t}(S) < (1 - \frac{\epsilon}{\lambda})\,\mathbb{E}[f_{\mathsf{cv},t}(S)]\right]$$

$$= \Pr\left[\sum_{v \in V_t} x_{t,v,S} < (1 - \frac{\epsilon}{\lambda}) \sum_{v \in V_t} \mathbb{E}[x_{t,v,S}]\right].$$

It suffices to consider the case $\epsilon < \lambda$. The multiplicative form Chernoff bound implies

$$\Pr\left[\sum_{v \in V_t} x_{t,v,S} < (1 - \frac{\epsilon}{\lambda}) \sum_{v \in V_t} \mathbb{E}[x_{t,v,S]}]\right] \le \exp\left(-\epsilon^2 \sum_{v \in V_t} \mathbb{E}[x_{t,v,S]}/2\lambda^2\right)$$

$$\le \exp(\epsilon^2(1 - \epsilon)R/2) \le \frac{\delta}{n^{ck/4}}.$$

The second step follows from Eq. (7). We conclude the proof here. $\qquad\square$

Lemma 3.10 indicates pointwise approximation is sufficient to carry over the approximation ratio between two problems. We provide a proof here.

*Proof of Lemma 3.10.* By an union bound over all sets of size at most $k$, we know that

$$\Pr_{g \sim D}[\exists S, |S| \le k, |f(S) - g(S)| - \gamma] \le n^k \frac{\delta}{n^{ck}} \le \frac{\delta}{n^{ck/2}}.$$

Under the above event, we know that

$$f(S_g) \ge g(S_g) - \delta \ge \alpha \max_{S \subseteq V, |S| \le k} g(S) - \gamma \ge \alpha \max_{S \subseteq V, |S| \le k} f(S) - 2\gamma.$$

This concludes the proof. $\qquad\square$

Given an influence graph, let $\mathrm{AVG} \cdot m$ be the expected number of steps taken by random sampling a RR set, and let OPT be the maximum (expected) influence spread of a seed set of size at most $k$, then one has

**Lemma B.1** (Claim 3.3 in [13]). $\mathrm{AVG} \le \frac{\mathrm{OPT}}{n}$ .

We provide a proof for completeness.

*Proof.* Given a node $v$ and an edge $(u, w)$, let $x_{v,e} = 1$ if the edge $e$ is checked when one samples the RR set of node $v$. Then we have

$$\mathrm{AVG} \cdot m = \frac{1}{n} \sum_{v \in V} \sum_{e \in E} \mathbb{E}[x_{v,e}] = \frac{1}{n} \sum_{e=(u,w) \in E} \mathbb{E}[|v : w \in R_v|].$$

Hence, we have

$$\mathrm{AVG} \cdot m = \frac{1}{n} \sum_{e=(u,w) \in E} \mathbb{E}[|v : w \in R_v|] = \frac{1}{n} \sum_{e=(u,v) \in E} \sigma(v) \le \frac{1}{n} \sum_{e=(u,v) \in E} \mathrm{OPT} = \frac{m}{n}\,\mathrm{OPT}.$$

We conclude the proof. $\qquad\square$

We provide proof for Lemma 3.11, which asserts with high probability, there are at most $O(\log n)$ iterations within a phase.

*Proof of Lemma 3.11.* For any $t \ge 0$, let $\mathsf{n}_t, \mathsf{m}_t$ be the number of nodes and edges at the beginning the $t$-th iteration, and let $\mathrm{AVG}_t \cdot \mathsf{m}_t$ be the average steps of sampling a random RR set. We can assume there is at least one edge in the graph, and therefore, $\mathrm{AVG}_0 \ge \frac{1}{\mathsf{n}_0 \mathsf{m}_0}$. Inside the phase, we must have $\mathsf{n}_0 \le \mathsf{n}_t \le 2\mathsf{n}_0$ and $\mathsf{m}_0 \le \mathsf{m}_t \le 2\mathsf{m}_0$ holds for any $t$. We prove that $\mathrm{AVG}_{t+1} \ge 2\,\mathrm{AVG}_t$

holds with high probability. Notice that $\text{AVG}_t \leq 1$, this means the algorithm restarts for at most $O(\log n)$ times. By Lemma 3.3, the sample size $K_t$ at the beginning of $t$-th iteration obeys

$$(1 - \epsilon) \cdot \frac{R}{\text{AVG}_t} \leq K_t \leq (1 + \epsilon) \cdot \frac{R}{\text{AVG}_t}. \tag{10}$$

with probability at least $1 - \frac{2\delta}{n^{ck/8}}$.

On the other side, for the $t$-th iteration, define $J_t, J_{t,1}, J_{t,2}$ similarly as Lemma 3.5. With probability at least $1 - \frac{2\delta}{n^{ck/8}}$, we have $J_t \in [J_{t,1}, J_{t,2}]$, and this indicates

$$8(1 - \epsilon)R \leq \sum_{j=1}^{J_{t,1}} \mathbb{E}[Z_j] \leq \sum_{j=1}^{J_t} \mathbb{E}[Z_j] \leq \sum_{j=1}^{J_{t,2}} \mathbb{E}[Z_j] \leq 8(1 + \epsilon)R. \tag{11}$$

The first and last step follow from the definition of $J_{t,1}$ and $J_{t,2}$, the second and third step follow from $J_t \in [J_{t,1}, J_{t,2}]$. Moreover, we also know that

$$\sum_{j=1}^{J_t} \mathbb{E}[Z_j] = \frac{K_t}{\mathsf{n}_t} \mathsf{n}_{t+1} \text{AVG}_{t+1}. \tag{12}$$

as we include the RR set of each $\mathsf{n}_t$ node with probability $\frac{K_t}{\mathsf{n}_t}$. Therefore, we have

$$2K_t \text{AVG}_{t+1} \geq \frac{K_t}{\mathsf{n}_t} \mathsf{n}_{t+1} \text{AVG}_{t+1} \geq 8(1 - \epsilon)R \geq \frac{8(1 - \epsilon)}{1 + \epsilon} K_t \text{AVG}_t \geq 4K_t \text{AVG}_t$$

The first step follows from $\mathsf{n}_{t+1} \leq 2\mathsf{n}_0 \leq 2\mathsf{n}_t$. The second step follows from Eq. (11) and Eq. (12). The third step comes from Eq. (10), and we assume $\epsilon < 1/3$ in the last step. Hence, we have proved with probability at least $1 - \frac{4\delta}{n^{ck/8}}$, $\text{AVG}_{t+1} \geq 2\text{AVG}_t$. Taking an union bound over $t$ and combining the fact that $\text{AVG}_0 \geq \frac{1}{\mathsf{n}_0 \mathsf{m}_0}$ and $\text{AVG}_t \leq 1$, we conclude with probability $1 - \frac{4\delta}{n^{ck/16}}$, the algorithm restarts at most $O(\log n)$ times within each phase. □

We wrap up the proof of Lemma 3.2

*Proof of Lemma 3.2.* We first prove the correctness of the algorithm. By Lemma 3.8 and Lemma 3.9, we know that at any time step $t$, the *normalized* coverage function $f_{\text{cv},t}$ defined on $\mathcal{H}_{\text{cv}}$ gives a good approximation on the influence spread function. In particular, we have that with probability at least $1 - \frac{4\delta}{n^{ck/8}}$, one has

$$|f_{\text{cv},t}(S) - \mathbb{E}[f_{\text{cv},t}(S)]| = |f_{\text{cv},t}(S) - \sigma_t(S)| \leq \epsilon \, \text{OPT}_t$$

Combining with Lemma 3.10, suppose we can solve the dynamic MAX-k coverage with approximation $\alpha$, then our algorithm maintains a solution set $S_t$ that satisfies

$$f(S) \geq (\alpha - 2\epsilon) \, \text{OPT}_t.$$

We next focus on the amortized running time. Since the number of edges and nodes can only doubled for most $O(\log n)$ times, there are $O(\log n)$ phases. While within one phase, by Lemma 3.11, with probability $1 - \frac{4\delta}{n^{ck/16}}$, our algorithm restarts for at most $O(\log n)$ times. Each time our algorithm restarts, it invokes the ESTIMATE procedure for once. This steps takes $Rm_0 = m_0 ck\epsilon^{-2} \log n$ steps in total and has $O(k\epsilon^{-2} \log n)$ amortized time per update. The algorithm constructs $\mathcal{H}_{\text{est}}$ and $\mathcal{H}_{\text{cv}}$, we calculate their cost separately. For constructing $\mathcal{H}_{\text{est}}$, our algorithm takes at most $16R_0 = 16m_0 ck\epsilon^{-2} \log n$ steps in total and has $O(k\epsilon^{-2} \log n)$ amortized time per update. For the construction of $\mathcal{H}_{\text{cv}}$, by Lemma 3.4 and Lemma 3.6, with probability at least $1 - \frac{9\delta}{n^{ck/8}}$, it takes less than $16(1 + \epsilon)^2 Rm_0 \leq 64m_0 ck\epsilon^{-2} \log n$ steps in total and $O(k\epsilon^{-2} \log n)$ amortized time per updates. Note that our algorithm not only needs to construct $\mathcal{H}_{\text{cv}}$, but also needs to maintains a set that has the (approximately) maximum coverage on $\mathcal{H}_{\text{cv}}$. This reduces to a dynamic MAX-k coverage problem, which by our assumption, can be solved in amortized running time of $\mathcal{T}_{\text{mk}}$. Taking an union bound over all steps and fix the constant $c$ to be greater than 24, we conclude with probability at least $1 - \delta$, the overall amortized running time per update is bounded by

$$\log n \cdot \log n \cdot (k\epsilon^{-2} \log n + k\epsilon^{-2} \log n + k\epsilon^{-2} \log n \cdot \mathcal{T}_{\text{mk}}) \leq O(\mathcal{T}_{\text{mk}} k\epsilon^{-2}(\log n)^3).$$

This concludes the proof. □

# C Missing proof from Section 3.2

Lemma 3.13 ensures the approximation guarantee of the algorithm, we provide a detailed proof here.

*Proof of Lemma 3.13.* Fix a time step $t$, let OPT denote the value of the optimal solution, i.e. $\text{OPT} = \max_{S,|S| \leq k} f_t(S)$. For ease of notation, we drop the subscript $t$ in the rest of the proof. There exists an index $i \in I$ such that

$$(1+\epsilon)^i = \text{OPT}_i \leq \text{OPT} < \text{OPT}_{i+1} = (1+\epsilon)^{i+1}.$$

We prove the $i$-th thread outputs a good solution set $S_i$.

First, suppose $|S_i| = k$. Let $s_{i,j}$ be the $j$-th element added to the set $S_i$, and denote $S_{i,j} = \{s_{i,1}, \ldots, s_{i,j}\}$, $j \in [k]$. Our algorithm guarantees that

$$f(S_{i,j+1}) - f(S_{i,j}) \geq \frac{1}{k}(\text{OPT}_i - f(S_{i,j}))$$

Then, we have that

$$\begin{aligned}
\text{OPT}_i - f(S_{i,k}) &= \text{OPT}_i - f(S_{i,k-1}) + f(S_{i,k-1}) - f(S_{i,k}) \\
&\leq \text{OPT}_i - f(S_{i,k-1}) - \frac{1}{k}(\text{OPT}_i - f(S_{i,k-1})) \\
&\leq \left(1 - \frac{1}{k}\right)(\text{OPT}_i - f(S_{i,k-1})) \\
&\vdots \\
&\leq \left(1 - \frac{1}{k}\right)^k (\text{OPT}_i - f(\emptyset)) \\
&= \left(1 - \frac{1}{k}\right)^k \text{OPT}_i,
\end{aligned}$$

and therefore,

$$\begin{aligned}
f(S_i) &\geq \left(1 - \left(1 - \frac{1}{k}\right)^k\right)\text{OPT}_i \geq \left(1 - \frac{1}{e}\right)\text{OPT}_i \\
&\geq \left(1 - \frac{1}{e}\right)(1+\epsilon)^{-1}\text{OPT} \geq \left(1 - \frac{1}{e} - \epsilon\right)\text{OPT}.
\end{aligned}$$

On the otherside, if $|S_i| < k$, then we prove $f(S_i) \geq \text{OPT}_i \geq (1+\epsilon)^{-1}\text{OPT}$. We prove by contradiction and assume $f(S_i) < \text{OPT}_i$ for now. Let the optimal solution be $O = \{o_1, \ldots, o_k\}$. Then we claim that

$$f_{S_i}(o) < \frac{1}{k}(\text{OPT}_i - f(S_i)) \tag{13}$$

holds for all $o \in O$. The reason is that (i) if $o \in S_i$, then $f_{S_i}(o) = 0 < \frac{1}{k}(\text{OPT}_i - f(S_i))$. If $o \notin O$, since $|S| < k$, the above is guaranteed by our algorithm. Hence, we have

$$\text{OPT} = f(O) \leq f(S_i) + f_{S_i}(O) \leq f(S_i) + \sum_{j=1}^{k} f_{S_i}(o_j) < f(S_i) + k \cdot \frac{1}{k}(\text{OPT}_i - f(S_i)) = \text{OPT}_i.$$

The second step holds by monotonicity, the third step holds by submodularity and the fourth step holds by Eq. (13). This comes to a contradiction. Hence, we proved $f(S_i) \geq (1 - 1/e - \epsilon)\text{OPT}$ in both cases. $\square$

We next prove Lemma 3.14, which analyses the amortized running time

*Proof of Lemma 3.14.* It suffices to prove that for each thread $i \in I$, the amortized running time is $O(1)$. We specify some implementation details. For any set $S$, let $N(S)$ denote the all neighbors of $S$. We maintain a set $V_R^i$ that includes all nodes covered by the current set $S_i$, i.e. $V_R^i = N(S_i)$ We also maintain a set $V_u^i$ for each node $u$, which contains all element covered by node $u$ in $V_R \backslash V_R^i$, i.e., $V_u^i = |N(u) \backslash V_R^i|$. Finally, we also maintain an order (on cardinality) over the set $V_u^i$. This is used in the REVOKE procedure, where we retrieve the node $u$ with the maximum $|V_u^i|$ and compare it with $\frac{\mathrm{OPT}_i - |V_R^i|}{k}$, We are going to prove that we can maintain these data structures and perform all necessary operations in $O(1)$ amortized time.

Let $P(t)$ denote the number of operations performed on the $i$-th thread up to time $t$. For each edge $e = (u, v)$, let $X_e$ denote whether the edge $e$ is covered by $V_R^i$, i.e. $X_e = 1$ if $v \in V_R^i$ and $X_e = 0$ otherwise. Similarly, for each node $v \in V_R$, let $Y_v$ denote whether node $v$ is included in $V_R^i$, i.e., $Y_v = 1$ if $v \in V_R^i$ and $Y_v = 0$ otherwise. For each node $u \in V_L$, let $Z_u$ denote whether node $u$ is included in $S_i$, i.e., $Y_v = 1$ if $v \in S_i$ and $Y_v = 0$ otherwise. Define the potential function $\Phi : t \rightarrow \mathbb{R}^+$:

$$\Phi(t) = 2|V_t| + 2|E_t| + 2 \sum_{e \in E_t} X_e + \sum_{v \in V_R} Y_v + 2 \sum_{v \in V_L} Z_u.$$

Our goal is to show $P(t) \leq \Phi(t)$. This is sufficient for our purpose as one can easily show $\Phi(t) \leq 5t$. We prove the claim by induction. The claim holds trivially for the base case $t = 0$. We next assume $t > 0$ and consider the time step $t$. If a new node arrives, then we have that $P(t) = P(t-1) + 1$. Since $|V_t| = |V_{t-1}| + 1$ and other terms of $\Phi$ won't decrease, we have $\Phi(t) \leq \Phi(t-1) + 1$. Suppose a new edge $e = (u, v)$ arrives. (1) If $v \in V_R^i$, that is, the node $v$ has already been covered. Then $P(t) = P(t-1) + 2$ since we don't perform any additional operations. We also have $\Phi(t) = \Phi(t-1) + 2$ as $|E_t| = |E_{t-1}| + 2$ and the other term remains unchanged. (2.1) If $v \notin V_R^i$ and $u \in S_i$, then we need to expand the set $V_t \leftarrow V_{t-1} \cup \{v\}$ (one unit operation), delete node $v$ from $V_u^i$ if $v \in V_u^i$ ($\sum_{v \in V_R} |V_u^i \cap \{u\}|$ operations) and maintains the order of $\{V_{v'}^i\}_{v' \in V_R}$ (at most $\sum_{v \in V_R} |V_v^i \cap \{u\}|$ operations). Meanwhile, we have $|E_t| = |E_{t-1}| + 1$ and the term $2 \sum_{e \in E_t} (1 - X_e)$ would increase for $2 \sum_{v \in V_R} |V_v^i \cap \{u\}|$, as these edges change from uncovered to covered. Hence, we still have $P(t) - P(t-1) \leq \Phi(t) - \Phi(t-1)$. (2.2) If $v \notin V_R^i$ and $u \notin S_i$. This may only cause two unit operations if there is no node $u$ satisfies $|V_u^i| \geq \frac{\mathrm{OPT}_i - |V_R^i|}{k}$. This time, we have $P_t = P_t + 2$ $\Phi(t) = \Phi(t-1) + 2$ as $|E_t| = |E_{t-1}| + 1$. On the other side, if there exists some node $u$ with large marginal. We need to add $u$ to $V_R^i$ (1 unit operation), add nodes in $V_u^i$ to $V_R^i$ ($|V_u^i|$ operations) and removes nodes in $V_u^i$ from all other set $V_{u'}^i$ ($\sum_{v \in V_r} |V_u^i \cap V_v^i|$ operations in total). We also want to maintain an order on $\{V_{v'}^i\}_{v' \in V_R}$, and this takes less than $\sum_{v \in V} |V_u^i \cap V_v^i|$ operations in total. Meanwhile, for the potential function, the term $2 \sum_{e \in E_t} X_t$ increases for $2 \sum_{v \in V} |V_u^i \cap V_v^i|$, as this number of edges change from uncovered to covered. The term $\sum_{v \in V_R} Y_v$ increases for ($|V_u^i|$ and term $2 \sum_{u \in V_L} Z_u$ will also increase by 2, as we augment the set $S_i$ by 1. Hence, we still have $P(t) - P(t-1) \leq \Phi(t) - \Phi(t-1)$ in this case. Finally, we note the that algorithm may call REVOKE multiple times upon the arrival of a new edge, and for each call, we call do perform similary analysis as (2.2). Hence, we conclude that $P(t+1) - P(t) \leq \Phi(t) - \Phi(t-1)$ holds for all $t$. We conclude the proof here. □

# D Missing proof from Section 4

*Proof of Theorem 4.3.* We assume $k = 1$ in our reduction. Let $m = n^{o(1)}$, $t = 2^{(\log n)^{1-o(1)}}$. Given an instance $\mathcal{A}, \mathcal{B}$ of the problem in Theorem 4.2, we reduce it to the dynamic influence maximization problem. We assume $|B_\tau| \geq t$ for all $\tau \in [n]$ as we can duplicate the ground element for $t$ times. Consider the following influence graph $G = (V, E, p)$, where the node set $V$ are partitioned into $V = V_1 \cup V_2 \cup V_3$. There are $n$ nodes in $V_1$, denoted as $v_{1,1}, \ldots, v_{1,n}$. Intuitively, the $i$-th node corresponds to the set $A_i \in \mathcal{A}$. The set $V_2$ contains $m$ nodes, denoted as $v_{2,1}, \ldots, v_{2,m}$. Intuitively, they correspond to the ground set $[m]$. For any node $v_{1,i} \in V_1$ and node $v_{2,j} \in V_2$, there is a directed edge from $v_{1,i}$ to $v_{2,j}$, iff the $j$-th element is contained in the set $A_1$. We associate the influence probability 1 to every edge between $V_1$ and $V_2$. The set $V_3$ contains $m^2 t$ nodes, denoted as $\{V_{3,j,\ell}\}_{j \in [m], \ell \in [mt]}$. There is a directed edge with influence probability 1 from node $v_{2,j}$ to $v_{3,j,\ell}$, for each $j \in [m], \ell \in [mt]$.

Consider the following update sequence of the DIM problem. The graph $G$ is loaded first and then followed by $n$ consecutive epochs. In the $\tau$-th epoch, all edges between $V_2$ and $V_3$ are deleted, and for each $j \in B_\tau$, we add back the edge between $v_{2,j}$ and $v_{3,j,\ell}$ for all $\ell \in [mt]$.

We first calculate the total number of updates. It takes $n + m + mt = n^{1+o(1)}$ steps to insert all nodes in $V$ and takes at most $mn + mt = n^{1+o(1)}$ to insert all edges in $E$. We delete/insert at most $m^2 t$ edges in each epoch, and since there are $n$ epochs, the total number operations are bounded by $nm^2 t = n^{1-o(1)}$. Hence, the total number of updates is at most $n^{1+o(1)}$.

Suppose on the contrary, there exists an algorithm for DIM problem that achieves $2/t$-approximation in $n^{1-\epsilon}$ time, we then derive a contradiction to SETH. Under the above reduction, we output YES, if for some epoch $\tau \in [n]$, the DIM algorithm outputs a solution with influence spread greater than $2m|B_\tau|$. We output NO otherwise. Note the influence of a node can be computed in $m = n^{o(1)}$ times.

**Completeness.** Suppose there exists $A_i \in \mathcal{A}$, $B_\tau \in \mathcal{B}$ such that $B_\tau \subseteq A_i$. Then in the $\tau$-th epoch, by taking node $v_{1,i}$ in the seed set, the influence spread at least $(mt+1)|B_\tau| + 1$. Since the DIM algorithm gives $2/t$-approximation, the influence is greater than $2m|B_\tau|$ in this case. Hence, we indeed output YES.

**Soundness.** Suppose $|A_i \cap B_\tau| < |B_\tau|/t$ for any $i, \tau \in [n]$, then we prove the influence spread is no more than $2m|B_\tau|$ for any epoch. This is clearly true for nodes in $V_2$ and $V_3$, as their influence is no more $mt + 1 < 2|B_\tau|m$. Here, we use the fact that $|B_\tau| \geq t$. For nodes in $V_1$, since the intersection of $A_i$ and $B_\tau$ is less than $|B_\tau|/t$, and a node $v_{2,j} \in V_2$ has influence $1 + mt$ if $j \in B_\tau$ and it has influence 1 otherwise. We conclude for any node $v_{1,i}$, its influence is at most

$$1 + m + \frac{1}{t}|B_\tau|mt = 1 + m + |B_\tau|m \leq 2|B_\tau|m.$$

Hence, we output NO in this case.

In summary, the reduced DIM requires $n^{1+o(1)}$ updates and queries and it gives an answer for the problem in Theorem 4.2. Hence, we conclude under SETH, there is no $2/t$-approximation algorithm unless the amortized running time is $n^{1-\epsilon}$. $\qquad\square$

*Proof of Theorem 4.4.* The reduction is similar to the one in Theorem 4.4. Let $m = n^{o(1)}$, $t = 2^{(\log n)^{1-o(1)}}$, $k = 1$. Given an instance $\mathcal{A}, \mathcal{B}$ of the problem in Theorem 4.2, we assume $|B_\tau| \geq t$ for all $\tau \in [n]$ Consider the following influence graph $G = (V, E, w)$, where the node set $V$ are partitioned into $V = V_1 \cup V_2 \cup V_3 \cup V_4$. There are $n$ nodes in $V_1$, denoted as $v_{1,1}, \ldots, v_{1,n}$ and there are $m$ nodes in $V_2$, denoted as $v_{2,1}, \ldots, v_{2,m}$. For any node $v_{1,i} \in V_1$ and node $v_{2,j} \in V_2$, there is a directed edge from $v_{1,i}$ to $v_{2,j}$, iff the $j$-th element is contained in the set $A_1$. We associate the weight to be $1/n$ to every edge between $V_1$ and $V_2$. The set $V_3$ contains $m^2 t$ nodes, denoted as $\{V_{3,j,\ell}\}_{j \in [m], \ell \in [mt]}$. There is a directed edge with weight 1 from node $v_{2,j}$ to $v_{3,j,\ell}$, for each $j \in [m], \ell \in [mt]$. The set $V_4$ contains $nm^2 t$ nodes, denoted as $\{V_{4,j,\ell,b}\}_{j \in [m], \ell \in [mt], b \in [n]}$. There is a directed edge with weight 1 from node $v_{3,j,\ell}$ to $v_{3,j,\ell,b}$, for each $j \in [m], \ell \in [mt], b \in [n]$. We assume the prescribed set is $V_1$, that is, we are only allowed to select seeds from $V_1$.

We use the same update sequence of the DIM problem. The graph $G$ is loaded first and then followed by $n$ consecutive epochs. In the $\tau$-th epoch, all edges between $V_2$ and $V_3$ are deleted, and for each $j \in B_\tau$, we add back the edge between $v_{2,j}$ and $v_{3,j,\ell}$ for all $\ell \in [mt]$.

The total number of updates is still at most $n^{1+o(1)}$, as the total number of edges between $V_3$ and $V_4$ is at most $nmt^2 = n^{1+o(1)}$ and we only insert them once. Suppose on the contrary, there exists an algorithm for DIM problem that achieves $2/t$-approximation in $n^{1-\epsilon}$ time, we then derive a contradiction to SETH. Under the above reduction, we output YES, if for some epoch $\tau \in [n]$, the DIM algorithm outputs a solution with influence spread greater than $2m|B_\tau|$. We output NO otherwise. Again, the influence of a node can be computed in $m = n^{o(1)}$ times.

**Completeness.** Suppose there exists $A_i \in \mathcal{A}$, $B_\tau \in \mathcal{B}$ such that $B_\tau \subseteq A_i$. Then in the $\tau$-th epoch, by taking node $v_{1,i}$ in the seed set, the influence spread at least $\frac{1}{n}|B_\tau| \cdot nmt + 1 = |B_\tau|mt + 1$. Since the DIM algorithm gives $2/t$-approximation, the influence is greater than $2m|B_\tau|$ in this case. Hence, we indeed output YES.

**Soundness**. Suppose $|A_i \cap B_\tau| < |B_\tau|/t$ for any $i, \tau \in [n]$, then we prove that no node in $V_1$ has influence spread more than $2m|B_\tau|$, in any epoch. Since the intersection of $A_i$ and $B_\tau$ is less than $|B_\tau|/t$, and a node $v_{2,j} \in V_2$ has influence $1 + mt + mtn$ if $j \in B_\tau$ and it has influence 1 otherwise. We conclude for any node $v_{1,i}$, its influence is at most

$$1 + m + \frac{1}{t}|B_\tau| \cdot \frac{1}{n}(1 + mt + nmt) < 2|B_\tau|m.$$

Hence, we output NO in this case.

In summary, the reduced DIM requires $n^{1+o(1)}$ updates and queries, and it gives an answer for the problem in Theorem 4.2. Hence, we conclude under SETH, there is no $2/t$-approximation algorithm unless the amortized running time is $n^{1-\epsilon}$. $\qquad\square$