# OpenReview forum: "Dynamic influence maximization"
_NeurIPS.cc/2021/Conference — NeurIPS 2021 Poster_

### Official Review · Reviewer_fcLb · 2021-07-15

**Rating:** 7
**Confidence:** 4

**Summary:**

This paper studies influence maximization (IM) in a dynamically evolving social network, where at each time step, one node/edge may emerge or disappear. The authors consider two evolution models: the incremental model (only node/edge insertion is allowed; e.g., the DBLP network) and the full model (both insertion and deletion are allowed; e.g., the Twitter network).

Under the incremental model, a $(1-1/e-\epsilon)$-approximation algorithm is proposed with $k\cdot \textrm{poly}(\log n, \epsilon^{-1})$ amortized running time. This is an algorithmic breakthrough because previous dynamic IM algorithms [44, 36, 20] all need $\Omega(n)$ amortized running time in the worst cases.

Under the full model, it is proved that no algorithm can achieve a meaningful approximation ratio within $O(n^{1-o(1)})$ amortized running time under the Strong Exponential Time Hypothesis.

Both theoretical results are proved under the classical IC and LT models.

**Limitations And Societal Impact:**

Please see the main review.

**Main Review:**

The theoretical results in this paper are novel from my perspective. The used techniques are also different from previous dynamic IM studies and dynamic submodular optimization studies. I did not check the proof correctness in the supplementary material, but the logic and intuition explained in the main paper seem to be smooth.

A few questions and concerns:
- You claim that several previous studies on dynamic IM [44, 36, 20] need $\Omega(n)$ amortized running time in the worst cases. Could you summarize their accurate running time in a table? Also, do they bear the same setting as yours? Do they allow node/edge deletion?

- The theoretical results in this paper are novel and inspiring, but I still feel empirical evaluation is needed. Unlike the general submodular optimization problem, IM is a practical task. Under the incremental model, I would suggest comparing the empirical running time and spread of the proposed model and previous studies [44, 36, 20] if they are applicable in this setting.

- In Lines 89-95, you mention the difference between your results and recent studies on dynamic submodular maximization. Specifically, their results are measured in query complexity. I feel this difference is not essential because one can multiply their complexity by the complexity of computing each query (i.e., the complexity of computing the spread given the seed set; can be accelerated using RR sets) when applying their approach to dynamic IM. Maybe the more essential difference is that in dynamic IM, $\sigma(\cdot)$ is changing when nodes/edges are inserted and deleted?

- There are many typos in the paper. I would suggest the authors carefully check their writings again. To give some examples,
Line 49: maintains -> maintain. Line 64: seminar -> seminal. Line 147: that is say -> that is to say.

**Time Spent Reviewing:**

4

---

> ### Author Response · Authors · 2021-08-08
> **Reply to Reviewer fcLb**
>
> We thank the reviewer for the insightful comments! We address the questions below.
>
> “You claim that several previous studies on dynamic IM [44, 36, 20] need $\Omega(n)$ amortized running time in the worst cases. Could you summarize their accurate running time in a table? Also, do they bear the same setting as yours? Do they allow node/edge deletion?”
>
> We summarize the exact running time of these papers below.
>
> [20, 36] allow both node/edge deletion. The time complexity of [20] is $\Omega(kn)$ and achieves $1/2$-approximation, the time complexity of [36] is $\Omega(\epsilon^{-3} (m+n) \log n)$ and achieves $(1-1/e)$ approximation.
>
> [44] considers the sliding window model, where only the most recent incremental changes are considered. Hence, they are more close to the incremental model (but with a moderate generalization). The time complexity of [44] is $\Omega(\epsilon^{-2}n\log n\log k)$ and only gives $(1/4 -\epsilon)$ approximation.
>
> “The theoretical results in this paper are novel and inspiring, but I still feel empirical evaluation is needed. Unlike the general submodular optimization problem, IM is a practical task. Under the incremental model, I would suggest comparing the empirical running time and spread of the proposed model and previous studies [44, 36, 20] if they are applicable in this setting.”
>
> Thanks for the suggestion! We will consider adding experiments in the camera ready version if the paper gets accepted.
>
> “In Lines 89-95, you mention the difference between your results and recent studies on dynamic submodular maximization. Specifically, their results are measured in query complexity. I feel this difference is not essential because one can multiply their complexity by the complexity of computing each query (i.e., the complexity of computing the spread given the seed set; can be accelerated using RR sets) when applying their approach to dynamic IM. Maybe the most essential difference is that in dynamic IM, $\sigma(\cdot)$ is changing when nodes/edges are inserted and deleted?”
>
> Thanks for pointing this out. One can run a query algorithm and measure the computation cost by multiplying the query complexity with the time of computing each query. However, for dynamic influence maximization, the time complexity of computing a query $\sigma(\cdot)$ is $\Omega(m)$, which is inherently large. So in general, one should avoid directly applying the algorithm under the query model.
>
> To make this more clear, note that there exists 1/2-approximation algorithm for (fully) dynamic submodular maximization problem that uses polylogarithmic number of queries per update,  but for DIM problem, we prove that one can not achieve any constant approximation with $n^{1-o(1)}$ amortized running time.
>
> “There are many typos in the paper. I would suggest the authors carefully check their writings again. To give some examples, Line 49: maintains -> maintain. Line 64: seminar -> seminal. Line 147: that is say -> that is to say.”
>
> Thanks for pointing this out, we will correct them and carefully check the writings.

---

### Official Review · Reviewer_isgZ · 2021-07-15

**Rating:** 6
**Confidence:** 2

**Summary:**

The paper presents new algorithms and hardness results for dynamic influence maximization, where dynamics come from one of two models: (1) the incremental model, which consider only the addition of new nodes and edges in a graph, and (2) the fully dynamic model, where nodes and edges can also be removed. Previous work has considered only heuristics that satisfy certain approximation guarantees but do not come with good amortized cost for updating the see set in the dynamic case. This paper provides a (1-1/e-eps) approximation for the optimal solution with k*poly(log n, 1/eps) approximation for the incremental model. For the fully dynamic model, the paper presents a hardness of approximation result if the Strong Exponential Time Hypothesis is true.

The proof for the approximation algorithm for the incremental model is based on two major pieces. First, the paper provides a reduction from dynamic influence maximization to dynamic MAX-k coverage. Then, a new algorithm for dynamic Max-k coverage is given. This follows a previous strategy for the static version of the problem, though a significant amount of theoretical work is required to extend both parts to the dynamic case.

The work is primarily theoretical and no implementations or experimental results are given.


**Limitations And Societal Impact:**

Yes

**Main Review:**

Overall, the contributions of the paper substantially extend previous work on influence maximization in static graphs, and the paper involves a significant theoretical effort. However, the theoretical details are quite hard to parse in many places and it's hard to tell what the algorithm is actually doing and hard to verify its correctness. This is no doubt at least partially due to my own unfamiliarity with some of the related work, as well as to the inherent complexity of making key techniques for influence maximization work in a dynamic setting. Nevertheless, even in the technical overview there are a number of statements that seem overly confusing, as well as a number of technical terms and ideas that are used without proper definition or explanation. I list several examples below, in the hopes that a re-write might make this more accessible to a wider audience.

* The word "steps" is being used to mean many different things in explanations on page 4 and following. First, the graph is dynamically changing at different time *steps* t. But then in the technique overview (starting line 175), I believe "steps" also starts to be used in reference to steps taken by a random instantiation of the process for finding a reverse reachable set (though I'm not sure what a step is here!). Then "steps" is used to refer to the two stages of the algorithm (line 180). I'd recommend the authors take some time to re-write the technical overview and try to distinguish more carefully between steps/stages, and concretely what a step means in each setting. This may seem minor, but without clarification and some more technical precision, it's not clear to me what is going on even in the overview, let alone the pages following it.
* The idea of a reverse reachable set is clear in the static setting (Definition 2.1), but it's not clear to me what a Reverse Reaching Set should mean in the dynamic setting or what it might mean to sample a reverse reachable set when nodes and edges are changing. These concepts are used repeatedly, but there is no concrete explanation of what this means. A more precise explanation might help here too.
* Related to both of the above points, when we get to Algorithm 2,  what notion of "steps" is being used here and what does it means to sample an RR set?
* The definition of dynamic influence maximization at the top of page 4 to me is clear. However, this relies on some notion of a time step t, but we don't see any reference to any time step t throughout the technical overview. It's unclear what steps are happening at different t values. For example, how does this factor into what it means to sample an RR set? Does sampling an RR set somehow happen across time steps t? (I'm not sure how it could, but I can't tell!). Does sampling RR sets happen at each time step? If so, how does that work?

Overall, I'd recommend a re-write of at least the technical overview on page 4 and 5, with some added technical precision. The main results seem strong and the very high level overview of the approach makes sense (i.e., reduction from DIM to dynamic MAX-k coverage + new algorithm for dynamic MAX-k coverage), but beyond this I'm unable to parse what is actually going on in the technical details.

**Time Spent Reviewing:**

2.5

---

> ### Author Response · Authors · 2021-08-08
> **Clarifying the presentation issues**
>
> We thank the reviewer for the thoughtful comments!
>
> We believe most of the issues mentioned in this review are related to presentations. We promise to revise the paper and better convey the idea in the camera ready version if the paper gets accepted.
>
>
>
> “The word "steps" is being used to mean many different things in explanations on page 4 and following. First, the graph is dynamically changing at different time steps t. But then in the technique overview (starting line 175), I believe "steps" also starts to be used in reference to steps taken by a random instantiation of the process for finding a reverse reachable set (though I'm not sure what a step is here!). Then "steps" is used to refer to the two stages of the algorithm (line 180). I'd recommend the authors take some time to re-write the technical overview and try to distinguish more carefully between steps/stages, and concretely what a step means in each setting. This may seem minor, but without clarification and some more technical precision, it's not clear to me what is going on even in the overview, let alone the pages following it.”
>
> Sorry for the confusion. Your interpretation is mostly correct. We will make this precise.
>
> “The idea of a reverse reachable set is clear in the static setting (Definition 2.1), but it's not clear to me what a Reverse Reaching Set should mean in the dynamic setting or what it might mean to sample a reverse reachable set when nodes and edges are changing. These concepts are used repeatedly, but there is no concrete explanation of what this means. A more precise explanation might help here too.”
>
> Thanks for pointing this out. In the dynamic settings, a Reverse Reachable Set is being dynamically maintained (e.g. see Algorithm 5). But as noted by the second reviewer, the description of Algorithm 5 is quite high level and some implementation details have been omitted. We will make them clear in the later version.
>
> “Related to both of the above points, when we get to Algorithm 2, what notion of "steps" is being used here and what does it means to sample an RR set?”
>
> Here, the “step” means the total number of computation steps, or equivalently, the total number of edges that has been sampled in the RR sampling process.
>
> “The definition of dynamic influence maximization at the top of page 4 to me is clear. However, this relies on some notion of a time step t, but we don't see any reference to any time step t throughout the technical overview. It's unclear what steps are happening at different t values. For example, how does this factor into what it means to sample an RR set? Does sampling an RR set somehow happen across time steps t? (I'm not sure how it could, but I can't tell!). Does sampling RR sets happen at each time step? If so, how does that work?”
>
> Sorry for the confusion. The algorithm does not depend on the specific time step t. For example, when inserting a new coming edge (u,v), we only invoke Algorithm 5, which augments RR sets and the two coverage graphs $H_{est}, H_{cv}$, so $t$ is not an input parameter to our algorithm.
>
> "Overall, I'd recommend a re-write of at least the technical overview on page 4 and 5, with some added technical precision."
>
> Thanks for the suggestion, we will do it in the later version.
>
> Overall, our paper is built upon existing technique of IM algorithms  and combined with several new ideas, it is quite technically compact. But we would make them more clear to the general audience based on your suggestions. Thanks!

---

### Official Review · Reviewer_YAX6 · 2021-07-16

**Rating:** 8
**Confidence:** 4

**Summary:**

The paper provides a theoretical study on the dynamic influence maximization problem, that is finding the top k seed nodes in a social network that has the largest influence spread while the social network is changing (nodes and edges could be added or deleted over time). The paper considers two dynamic models: the incremental model where only node and edge additions are allowed, and fully dynamic model, where both additions and deletions are allowed. For the incremental model, the paper provides an algorithm that guarantees finding an approximately optimal seed set solution for all networks during the network evolution, while the amortized update cost is much smaller than the re-computation cost at every time step. For the fully dynamic model, the authors provide a strong negative result, showing that under a commonly adopted complexity assumption SETH, the fully dynamic influence maximization problem cannot be solved, more specifically, to achieve even slightly better than a nontrivial 1/n approximation ratio, one need almost linear amortized running time.

The contribution of the paper is on the systematic study on dynamic influence maximization. From the theory side, it is much better than existing studies on dynamic influence maximization, which is mostly heuristic based and does not provide speedup in amortized running time.

**Ethical Concerns:**

No ethical concerns.

**Limitations And Societal Impact:**

The work is purely theoretical, and no perceived negative societal impact exists.

**Main Review:**

The strength of the paper is its theoretical contribution to the dynamic influence maximization problem. This is the first study I see that provide strong theoretical results on two important dynamic models. On the incremental model, it provides the first theoretical algorithm that has significant speedup in terms of the amortized running time, where the existing studies usually do not provide any theoretical results and any speedup in amortized running time. To achieve this theoretical result, the proposed algorithm is nontrivial, and it combines several techniques to address several issues that arises due to the dynamic changes in the network. For the fully dynamic model, it provides a strong inapproximation result, which is the first such hardness result and also provides inside into the hardness of this setting.

In general, the paper has good original results, built upon recent research on dynamic submodular maximization and certain hardness results. It would be very useful for the community to understand these theoretical results, and since dynamic setting is very realistic, this could be important for practical considerations too. The paper also tries to explain the algorithm and proof ideas in a clean way, and in general the paper is well structured.

The weakness of the paper is that the key algorithm explanations have several unclear points and typos, making it hard to fully understand the algorithm and its novelty. I will list these issues below in the detailed comment section. I believe most of these issues can be addressed in the next revision, and I am looking forward to the authors' feedback to answer these issues and clean up the presentation. I did not check the details of the analysis, but has reasonable confidence that the martingale based analysis is mainly correct.


Detailed comments on the presentation issues:

- Line 12. For the hardness result, the approximation ratio uses the expression that is greater than 1, while for the algorithm, it uses the approximation ratio of 1-1/e, which is less than 1. This should be made consistent.

- Line 99. It uses V and E to denote the set of nodes and edges, and their size to be n and m. Since the network is dynamic, the authors should clearly state what V, E, n and m means. I believe it means the largest possible set of nodes and edges and their values. Please clarify. Also, later V is also used as the instant node set, and that could be confusing. I suggest that the authors always use $V_t$ to represent the current node set at some time t.

- Line 169-170. It says the amortized running time matches the state-of-the-art offline algorithm up to poly-logarithmic factors. This statement is unclear to me. If we totally have T updates, the total running time of their dynamic algorithm should be much faster than re-run the offline algorithm at each update. The state-of-the-art offline algorithm for each update is in time $O(k(n+m) \log n/\epsilon^2)$. So what does it mean by matching the algorithm? I just realized the authors perhaps mean by growing a graph from an empty graph in n+m steps, their algorithm would achieve almost the same running time. Please explain this in a better way to convey the idea that their algorithm is much faster than re-run the offline algorithm at each update.

- Line 220. $OPT_i$ is not explained in the context, should directly say it is $(1+\epsilon)^i$.

- Section 3.1. The explanation of the algorithm in the text as well as the algorithm pseudocode should be improved. There are number of points that are unclear to me in the first reading. The following is a list of such issues.

- Line 240. Variable $R$ and $m_0$ are not explained in the context. One has to read the pseudocode to get some idea.

- Algorithm 1, line 2, notation V and E is better changed to dynamic notations, such as V_t or E_t, to avoid confusing with the largest node and edge set notation. For example, in line 3 of Algorithm 2, when it says uniformly over V, I initially thought it is uniformly over the largest node set, but I guess it means uniform over the current node set V.

- Algorithm 2, line 5, "step" is a technical term that is not explained precisely. What is exactly a step? Moreover, in earlier description (see line 131), it refers to one time step t as the step for adding a node or edge (or delete a node or edge). I believe these two steps are different, but the authors do not provide clarification. I suggest that for network change, we may call it round, and in each round, there is one network change. The time step is reserved for computational steps within a round. Please specify what one step really means. Is it that randomly selecting one reverse edge in the RR set generation process one step?

- Algorithm 3, line 2. Constant c is not explained. What is its purpose?

- Algorithm 3, line 4. This is to set a probability p. What if $K > n_0$ and the $p$ is not a probability? I never see an explanation here. Actually, I believe this is a very important point. If $p < 1$ and it is indeed a probability, then line 5-11 only samples one RR set for each node $v$. Is it enough? In the RIS approach for the classical IM problem, it is possible that on average each node $v$ would have many RR set samples, otherwise, the estimation may not be accurate. Therefore, I feel that there is some gap here. Is it possible that $K > n_0$ and this suggests that each node $v$ should have multiple RR set samples?

- Algorithm 3, line 9, INSERT-EDGE-COV(u,v), this routine is not explained in Section 3.1 and only explained in Section 3.2, but it is important to understand this routine in order to understand Algorithm 1-5.

- Algorithm 4, line 6. Notation $V_{L,est}$ is not explained, so do the notations in the next three lines. I have to guess that it means the left-side nodes in the bipartite graph H_{est}. Please clarify. Moreover, the left-side of H_{est} should be the nodes while the right-side of H_{est} should be RR sets, so technically, for line 8 and 9, it should be RR sets $\{u\}$ with a single node, not the single node $u$ itself.

- Algorithm 5, line 7 and line 15. There is no explanation of what these augmentation operations do. Please clarify. This is actually the important incremental step of the model. Is it just simply revisiting all the RR sets to consider adding the edge $(u,v)$? How to control the time complexity? This time complexity is also related to the time step estimation in line 9.

- Algorithm 5, line 15 and 16. Why the subscript est appears in the notations? Is it a typo, and should it be cv? Also, for all $v$ \in R^{new}_{v,cv} \setminus R_{v,cv} looks strange to me, since $v$ appears both on the left and as the subscripts in the right-hand side.

- Line 306. It is strange to return a value. I think it should return argmax among $S_i$, where $i \in I$.

- Algorithm 6 to 8. Function $f$ is used in these algorithms. What is exactly the $f$? Should it be $f_{cv}$ derived from the coverage graph $\mathcal{H}_{cv}$?

- Line 490. "Xi Chen" should be "Wei Chen"

Overall, the above listed issues are mostly presentation issues. If the authors can satisfactorily address all these issues in the rebuttal, my rating on the paper could be further increased.

There are also a few other studies on dynamic influence maximization. Although they are not directly related to the theoretical study of the current paper, they are early studies in this area that should be cited:

- Aggarwal C, Lin S, and Yu P S. On influential node discovery in dynamic social networks. In Proceedings of SIAM International Conference on Data Mining (SDM), Anaheim, USA, Apr 2012

- Liu X, Liao X, Li S, Zhang J, Shao L, Huang C, and Xiao L. On the shoulders of giants: incremental influence maximization in evolving social networks. Complexity 2017

============ Post rebuttal comments ==================

The authors feedback has addressed my comments and concerns, which are mostly on the presentation and clarity of the paper. In particular, the treatment of p in Algorithm 3 is clear now --- it is not just a probability, and could allow multiple samples.


**Time Spent Reviewing:**

4 hours

---

> ### Author Response · Authors · 2021-08-08
> **Clarifying the presentation issues**
>
> We thank the reviewer for the insightful comments. These are really helpful for improving the presentation of this paper!
>
> Most of the concerns are related to the presentation issue and most of the time, your understanding is correct. Please see the response below for detailed explanation.
>
> “1. Line 12. For the hardness result, the approximation ratio uses the expression that is greater than 1, while for the algorithm, it uses the approximation ratio of 1-1/e, which is less than 1. This should be made consistent.”
>
> Thanks for pointing this out, we will make them consistent.
>
>
> “2. Line 99. It uses V and E to denote the set of nodes and edges, and their size to be n and m. Since the network is dynamic, the authors should clearly state what V, E, n and m means. I believe it means the largest possible set of nodes and edges and their values. Please clarify. Also, later V is also used as the instant node set, and that could be confusing. I suggest that the authors always use V_t to represent the current node set at some time t. ”
>
> Thanks for pointing these out, we will make the notation consistent in the camera ready version if the paper gets accepted. Here, V, E, n, m indeed mean the largest possible set of nodes and edges and their values.
>
>
> “3. Line 169-170. It says the amortized running time matches the state-of-the-art offline algorithm up to poly-logarithmic factors. This statement is unclear to me. If we totally have T updates, the total running time of their dynamic algorithm should be much faster than re-run the offline algorithm at each update. The state-of-the-art offline algorithm for each update is in time $O(k(n+m)\log⁡ n \cdot \epsilon^{-2})$. So what does it mean by matching the algorithm? I just realized the authors perhaps mean by growing a graph from an empty graph in n+m steps, their algorithm would achieve almost the same running time. Please explain this in a better way to convey the idea that their algorithm is much faster than re-run the offline algorithm at each update.”
>
> Thanks for pointing this out, your understanding is correct. It is common to use the term “matching state-of-the-art offline algorithm” in the TCS literature on dynamic algorithms, but we definitely agree it could be confusing for the general audience. We will explain it in better ways.
>
> “4. Line 240. Variable R and $m_0$ are not explained in the context. One has to read the pseudocode to get some idea."
>
> "5. Algorithm 1, line 2, notation V and E is better changed to dynamic notations, such as $V_t$ or $E_t$, to avoid confusing with the largest node and edge set notation. For example, in line 3 of Algorithm 2, when it says uniformly over V, I initially thought it is uniformly over the largest node set, but I guess it means uniform over the current node set V.”
>
> Thanks for the suggestion! Your interpretation is correct.
>
> “6. Algorithm 2, line 5, "step" is a technical term that is not explained precisely. What is exactly a step? Moreover, in earlier description (see line 131), it refers to one time step t as the step for adding a node or edge (or delete a node or edge). I believe these two steps are different, but the authors do not provide clarification. I suggest that for network change, we may call it round, and in each round, there is one network change. The time step is reserved for computational steps within a round. Please specify what one step really means. Is it by randomly selecting one reverse edge in the RR set generation process in one step?”
>
> Thanks for the suggestion! These two “steps” have different semantic meanings and your interpretation is correct, i.e., for the one in Algorithm 2, line 3, it means randomly selecting one reverse edge in the RR set generation process.
>
> “7. Algorithm 3, line 2. Constant c is not explained. What is its purpose?”
> The constant c is for bounding the failing probability, e.g. see Lemma 3.5.
>
> “8. Algorithm 3, line 4. This is to set a probability p. What if $K>n_0$ and the p is not a probability? I never see an explanation here. Actually, I believe this is a very important point. If $p<1$ and it is indeed a probability, then line 5-11 only samples one RR set for each node v. Is it enough? In the RIS approach for the classical IM problem, it is possible that on average each node v would have many RR set samples, otherwise, the estimation may not be accurate. Therefore, I feel that there is some gap here. Is it possible that $K>n_0$ and this suggests that each node v should have multiple RR set samples?
>
> Thanks for pointing this out! We indeed miss some details here, and apologize for this. When $K > n_0$, i.e., $p > 1$, we would sample multiple RR sets according to p. For example, say p = 3.5, we would sample each node 3 times with prob. ½ and 4 times with prob. ½.
>
> “9. Algorithm 3, line 9, INSERT-EDGE-COV(u,v), this routine is not explained in Section 3.1 and only explained in Section 3.2, but it is important to understand this routine in order to understand Algorithm 1-5. ”
>
> Thanks! We will seek better ways to explain this routine and make it clear.
>
> "10. Algorithm 4, line 6. Notation $V_L$,est is not explained, so do the notations in the next three lines. I have to guess that it means the left-side nodes in the bipartite graph $H_{est}$. Please clarify. Moreover, the left-side of $H_{est}$ should be the nodes while the right-side of $H_{est}$ should be RR sets, so technically, for line 8 and 9, it should be RR sets u with a single node, not the single node u itself."
>
> Your understanding of the notation is correct.
>
> “11. Algorithm 5, line 7 and line 15. There is no explanation of what these augmentation operations do. Please clarify. This is actually the important incremental step of the model. Is it just simply revisiting all the RR sets to consider adding the edge (u,v)? How to control the time complexity? This time complexity is also related to the time step estimation in line 9. ”
>
> We augment RR sets in this step and the details are explained below. We would consider adding the edge (u,v) to all RR sets, and when it is added to a RR set, we continue augmenting this RR set, i.e. sampling incoming edges of node u and so on. To control the time complexity, we maintain a double-linked list from any RR set to its nodes (i.e. all node it can reach), this step can double the computation cost at most twice; and when we consider adding edge (u,v), we don’t really check all RR sets, but only check the double-linked list and find all RR sets that include node v.
>
> Furthermore, when doing the estimation in line 9, we only consider the total number of steps on augmenting RR set (i.e., we do not include the step on maintaining the double-linked list and checking the list, though these are up to a constant factor).
>
> We apologize for not explaining the implementation details in the paper, and we would include it in the camera ready version if the paper gets accepted.
>
> “12. Algorithm 5, line 15 and 16. Why the subscript est appears in the notations? Is it a typo, and should it be cv? Also, for all $v \in  R_{v, cv}^{new}$,$ \backslash R_{v,cv}$ looks strange to me, since v appears both on the left and as the subscripts in the right-hand side."
>
> Thanks for pointing this out. Sorry, there are some typos around line 14-18. First, we should change all est to cv. Second, v should not appear on both sides of line 16, and the left side should be u instead of v. Third, we mix up notations for v around line 14-18, and it seems better to change most of v to v’. Sorry for this. In general, here we are trying to augment RR sets and the coverage graph $H_cv$.
>
> “13. Line 306. It is strange to return a value. I think it should return argmax among $S_i$, where $i\in I$.”
>
> Yes, it should return the set.
>
> “14. Algorithm 6 to 8. Function f is used in these algorithms. What is exactly the f? Should it be f_cv derived from the coverage graph Hcv?
>
> Yes, it is the coverage functions on $H_cv$.
>
>
> “15. Line 490. "Xi Chen" should be "Wei Chen"”.
>
> Oops, this is a typo, sorry for that.
>
> “16. There are also a few other studies on dynamic influence maximization. Although they are not directly related to the theoretical study of the current paper, they are early studies in this area that should be cited:
> Aggarwal C, Lin S, and Yu P S. On influential node discovery in dynamic social networks. In Proceedings of SIAM International Conference on Data Mining (SDM), Anaheim, USA, Apr 2012
> Liu X, Liao X, Li S, Zhang J, Shao L, Huang C, and Xiao L. On the shoulders of giants: incremental influence maximization in evolving social networks. Complexity 2017”
>
> Thanks again for pointing them out, we will cite them.

---

> > ### Comment · Reviewer_YAX6 · 2021-08-11
> > **confirm a response**
> >
> > Is there a typo in this sentence: " For example, say p = 3.5, we would sample each node 3 times with prob. ½ and 4 times with prob. ½."?
> >
> > Note that this is an important technical aspect of the algorithm that the authors did not make it clear, and I hope the authors could clarify. For the above example, should it be that we sample each node 3 times with probability 1, and sample it the 4th time with probability 1/2?

---

> > > ### Author Response · Authors · 2021-08-11
> > > **Re: confirm a response**
> > >
> > > Yes, when p=3.5, we would sample each node 3 times with probability 1 and sample it the 4th time with probability 1/2.

---

### Official Review · Reviewer_tdZf · 2021-07-27

**Rating:** 5
**Confidence:** 4

**Summary:**

The authors propose an efficient algorithm for dynamic influence maximization in the incremental setting where only nodes and edges in the graph can be added over time.  For the fully dynamic setting where the nodes and edges can be deleted as well, they give a hardness result.

**Ethical Concerns:**

None.

**Limitations And Societal Impact:**

None.

**Main Review:**

The paper is well-written and easy to follow.  Even as the results for the incremental version are interesting, the incremental setting does not accurately represent a social network.   The result of Wang et. al, 2017 gives a \eps (1 - \beta)/2 approximation by keeping O(log N/\beta) sampled nodes  in a sliding window of size N.  Even as the use of a sliding window in their paper suffers from the same drawback of supporting only additions to the network, they do present an algorithm that looks similar to the algorithm for the dynamic max-k coverage  in this paper.  A clearer comparison with the earlier work will help position this work.

Read the authors' feedback and updating my score slightly higher.  My concerns about the incremental setting still remain.

**Time Spent Reviewing:**

2

---

> ### Author Response · Authors · 2021-08-08
> **Addressing the concerns**
>
> We thank the reviewer for the helpful comments, these are useful at improving the presentation of this paper.
>
> However, we do feel the concerns raised by the reviewer are not sufficient to reject the paper. We respond to the two major concerns below.
>
> “Even as the results for the incremental version are interesting, the incremental setting does not accurately represent a social network.”
>
> We want to support our model and results from the following three perspectives.
>
> 1. Motivation. We do think incremental setting captures a wide range of social networks. For example, as we explain in our paper, social networks are involved incrementally under the well-established preferential attachment model [1], which is widely accepted and receives more than 5000 citations. Moreover, we give many examples of increments involving networks, including the DBLP and scientific co-authorship network.
>
> 2. Content of the paper. The content of this paper not only includes the algorithm for incremental setting, but also provides a strong negative result for the fully dynamic involving model. We believe considering the incremental setting is a natural and realistic approach to circumvent the hardness result.
>
> 3. Literature. There are many works in literature that considers incremental involving of social networks. For example, the work of Wang et al. 2017 (though the sliding window model is a slight generalization), as well as the work of [2] (thanks for the second reviewer for bringing this up to our attention). Hence, we do think incremental models are well established in the social network literature, not to say it is a fundamental model considered in dynamic algorithms literature.
>
>
> “Wang et. al, 2017 gives a \eps (1 - \beta)/2 approximation by keeping O(log N/\beta) sampled nodes in a sliding window of size N. Even as the use of a sliding window in their paper suffers from the same drawback of supporting only additions to the network, they do present an algorithm that looks similar to the algorithm for the dynamic max-k coverage in this paper. ”
>
> Thanks for pointing this out. Before we proceed to explain the difference, we do want to emphasize that we already cite the paper of Wang et al. 2017 in related work. More importantly, in the incremental setting, our algorithm achieves (1-1/e) approximation, whereas they achieve ¼ approximation; our amortized running time is $\tilde{O}(k)$ where their algorithm has running time at least $\tilde{\Omega}(n)$. Hence we give significant improvement over both running time and approximation ratio. Therefore, we don’t think this is a sufficient reason for the rejection.
>
> Now, we explain the details over the subroutine on dynamic max-k coverage and our technical improvement. For the dynamic max-k coverage subroutine, Wang et al. 2017 use “black box” reduction from streaming submodular maximization algorithm, which achieves 1⁄2 approximation by accepting element that has marginal contribution larger than $\frac{1}{2k}$OPT, and the approximation ratio is known to be tight by [3]. We circumvent the streaming lower bound with new ideas, and achieve 1-1/e approximation. Concretely, we dynamically maintain a threshold (i.e., not keeping it fixed, as explained in section 3.2), which could potentially take up to k value. The technical novelty is to analyze this new algorithm (outlined in Lemma 3.14) and prove its approximation guarantee (outlined in Lemma 3.13).
>
> We thank the reviewer for useful comments, and we hope the reviewer could reconsider the assessment on the quality of this paper. We do think it is a significant one in this subfield.
>
>
> [1] Liben‐Nowell, David, and Jon Kleinberg. "The link‐prediction problem for social networks." Journal of the American society for information science and technology 58.7 (2007): 1019-1031
>
> [2] Liu, Xiaodong, et al. "On the shoulders of giants: incremental influence maximization in evolving social networks." Complexity 2017 (2017).
>
> [3] Feldman, Moran, et al. "The one-way communication complexity of submodular maximization with applications to streaming and robustness." Proceedings of the 52nd Annual ACM SIGACT Symposium on Theory of Computing. 2020.

---

> > ### Comment · Reviewer_tdZf · 2021-08-27
> > **Post-feedback**
> >
> > Thanks for the clarifications.  I agree with your running time analysis.   I am still not convinced on the incremental setting is a natural model for many describing many social networks.  There are a non-trivial number of deletions as well.  Further, in addition to preferential attachment, homophily (number of common neighbors, graph distance) also plays an important role in the addition or deletion of an edge.  In fact, the paper you refer to [1] shows that preferential attachment performs poorly (Figure 3) compared to the other measures for link prediction in evolving networks.  Am I missing something?

---

> > > ### Author Response · Authors · 2021-08-27
> > > **Address concern**
> > >
> > > We thank the reviewer for detailed comments. We address the concerns below.
> > >
> > > "Further, in addition to preferential attachment, homophily (number of common neighbors, graph distance) also plays an important role in the addition or deletion of an edge. In fact, the paper you refer to [1] shows that preferential attachment performs poorly (Figure 3) compared to the other measures for link prediction in evolving networks".
> > >
> > > For the concern on  the preferential attachment model, we agree there are many other prediction models and preferential attachment might be too ideal for certain applications. However, we want to emphasize that
> > >
> > > 1. The preferential attachment model is well-accepted in the literature and therefore, it should be a reasonable motivation for our incremental model. Depending on the applications, there are also some supporting experiments on the model (e.g. see [2] in our previous response).
> > >
> > > 2. The preferential attachment model is only used as a motivation for our incremental evolving model, our algorithm applies to all incremental growing networks and thus certainly does not restrict to this model.
> > >
> > > 3. Last but not least, the influential paper [1] is trying to address the problem of link prediction, that is, predict future connection based on current network topology. This implicitly assumes the network is growing over time. In other words, though preferential attachment might not accurately predict the growth of the network, some other measurement could (e.g. common neighbors). This in turn indicates that the network is growing according to some network measurement.
> > >
> > >
> > > "I am still not convinced that the incremental setting is a natural model for many describing many social networks. There are a non-trivial number of deletions as well."
> > >
> > > For your general concerns on incremental models, we agree there could be deletions for some real world applications, and it could be a good future direction. However, we provide affirmative support for our model as follows.
> > >
> > > First of all, the paper not only considers the incremental model, we also provide a hardness result for the fully dynamic model. Based on recently developed machinery on fine grained complexity, we provide a highly non-trivial reduction from SETH (a commonly accepted complexity assumption), and prove that the fully dynamic IM problem is intractable, in the sense that it is impossible to achieve any constant approximation in o(n) time. This result essentially forbids any theoretically efficient algorithm for the general problem, and hence, this part alone already solves this long-standing problem. However, given dynamic IM is an important and practical relevant problem, we don’t want to stop at this point. A common way of circumventing theoretically hardness is to find suitable assumptions/models. Along this way, we come up with the incremental model. From our view, it is the most appropriate one based on the following three reasons.
> > >
> > > 1. It captures a large subset of applications. For example, the DBLP network (see the experiments of [4]) and scientific network.
> > >
> > > 2. It could be useful for general applications. For social networks like Facebook, Twitter, we agree there could be spot events that make local and temporal influence go up and down, but in the long term, the network follows a growing trend. Hence we believe our algorithm is certainly helpful in general applications.
> > > 3. It captures people’s belief on network evolving. As we said before, it captures many theoretical models on evolving networks (preferential attachment, common neighbor, etc,). Though these theoretical models are not perfect for real world applications, it at least supports that people believe social networks are generally growing.
> > >
> > >
> > > [4] Chen, Wei, Chi Wang, and Yajun Wang. "Scalable influence maximization for prevalent viral marketing in large-scale social networks." Proceedings of the 16th ACM SIGKDD international conference on Knowledge discovery and data mining. 2010.

---

### Decision · Program_Chairs · 2021-09-27

**Decision:**

Accept (Poster)

**Comment:**

The authors propose a new efficient algorithm for influence maximization in the incremental setting.  The theoretical contribution is interesting and novel and the problem is interesting and well-studied. The main limitation of the current write-up is its writing but that should be addressed by implementing the reviewers’ suggestions. Overall, the paper is a nice contribution and I suggest to accept it.